# Cell type-specific connectome predicts distributed working memory activity in the mouse brain

Xingyu Ding[1†], Sean Froudist-Walsh[1,2†], Jorge Jaramillo[1,3†], Junjie Jiang[1,4], Xiao-Jing Wang[1]*

[1]Center for Neural Science, New York University, New York, United States; [2]Bristol Computational Neuroscience Unit, School of Engineering Mathematics and Technology, University of Bristol, Bristol, United Kingdom; [3]Campus Institute for Dynamics of Biological Networks, University of Göttingen, Göttingen, Germany; [4]The Key Laboratory of Biomedical Information Engineering of Ministry of Education,Institute of Health and Rehabilitation Science,School of Life Science and Technology, Research Center for Brain-inspired Intelligence, Xi'an Jiaotong University, Xi'an, China

**\*For correspondence:**
xjwang@nyu.edu

[†]These authors contributed equally to this work

**Competing interest:** The authors declare that no competing interests exist.

**Abstract** Recent advances in connectomics and neurophysiology make it possible to probe whole-brain mechanisms of cognition and behavior. We developed a large-scale model of the multi-regional mouse brain for a cardinal cognitive function called working memory, the brain's ability to internally hold and process information without sensory input. The model is built on mesoscopic connectome data for interareal cortical connections and endowed with a macroscopic gradient of measured parvalbumin-expressing interneuron density. We found that working memory coding is distributed yet exhibits modularity; the spatial pattern of mnemonic representation is determined by long-range cell type-specific targeting and density of cell classes. Cell type-specific graph measures predict the activity patterns and a core subnetwork for memory maintenance. The model shows numerous attractor states, which are self-sustained internal states (each engaging a distinct subset of areas). This work provides a framework to interpret large-scale recordings of brain activity during cognition, while highlighting the need for cell type-specific connectomics.

## Editor's evaluation

This valuable modeling study helps to elucidate the conditions under which interactions across brain regions support working memory. Convincing evidence underscores the importance of not merely the strength and density of long-range connections, but also the cell type specificity of such connections, and solid results indicate that the density of inhibitory neurons, together with placement in a cortical hierarchy, plays a role in how strongly or weakly a brain region displays persistent activity. This work will be of interest to modelers studying the neural basis of working memory, as well as to neuroscientists interested in how global brain interactions shape the patterns of brain activity observed during working memory.

## Introduction

In contrast to our substantial knowledge of local neural computation, such as orientation selectivity in the primary visual cortex or the spatial map of grid cells in the medial entorhinal cortex, much less is understood about distributed processes in multiple interacting brain regions underlying

cognition and behavior. This has recently begun to change as advances in new technologies enable neuroscientists to probe neural activity at single-cell resolution and on a large-scale by electrical recording or calcium imaging of behaving animals (*Jun et al., 2017*; *Steinmetz et al., 2019*; *Stringer et al., 2019*; *Musall et al., 2019*; *Steinmetz et al., 2021*), ushering in a new era of neuroscience investigating distributed neural dynamics and brain functions (*Wang, 2022*).

To be specific, consider a core cognitive function called working memory, the ability to temporally maintain information in mind without external stimulation (*Baddeley, 2012*). Working memory has long been studied in neurophysiology using delay-dependent tasks, where stimulus-specific information must be stored in working memory across a short time period between a sensory input and a memory-guided behavioral response (*Fuster and Alexander, 1971*; *Funahashi et al., 1989*; *Goldman-Rakic, 1995*; *Wang, 2001*). Delay-period mnemonic persistent neural activity has been observed in multiple brain regions, suggesting a distributed working memory representation (*Suzuki and Gottlieb, 2013*; *Leavitt et al., 2017*; *Christophel et al., 2017*; *Xu, 2017*; *Dotson et al., 2018*, also see *Table 1*). Connectome-based computational models of the macaque cortex found that working memory activity depends on interareal connectivity (*Murray et al., 2017*; *Jaramillo et al., 2019*), macroscopic gradients of synaptic excitation (*Wang, 2020*; *Mejías and Wang, 2022*), and dopamine modulation (*Froudist-Walsh et al., 2021*). Mnemonic neural activity during a delay period is also distributed in the mouse brain (*Liu et al., 2014*; *Schmitt et al., 2017*; *Guo et al., 2017*; *Bolkan et al., 2017*; *Gilad et al., 2018*). New recording and imaging techniques as well as optogenetic methods for causal analysis (*Yizhar et al., 2011*), which are widely applicable to behaving mice, hold promise for elucidating the circuit mechanism of distributed brain functions in rodents.

Recurrent synaptic excitation represents a neural basis for the maintenance of persistent neural firing (*Goldman-Rakic, 1995*; *Amit, 1995*; *Wang, 2021*). In the monkey cortex, the number of spines (sites of excitatory synapses) per pyramidal cell increases along the cortical hierarchy, consistent with the idea that mnemonic persistent activity in association cortical areas including the prefrontal cortex is sustained by recurrent excitation that is stronger than in early sensory areas. Such a macroscopic gradient is lacking in the mouse cortex (*Gilman et al., 2017*; *Ballesteros-Yáñez et al., 2010*), raising the possibility that the brain mechanism for distributed working memory representations may be fundamentally different between mice and monkeys.

**Table 1.** Supplementary experimental evidence. The listed literature include experiments that provide supporting evidence for working memory activity in cortical and subcortical brain areas in the mouse or rat. These studies show either that a given area is involved in working memory tasks and/or exhibit delay period activity. Area name corresponds to what has been reported in the literature. Some areas do not correspond exactly to the names from the Allen common coordinate framework.

| Area | Supporting literature |
| --- | --- |
| ALM (MOs) (anterior lateral motor cortex) | *Guo et al., 2014*; *Li et al., 2016*; *Guo et al., 2017*; *Gilad et al., 2018*; *Inagaki et al., 2019*; *Wu et al., 2020*; *Voitov and Mrsic-Flogel, 2022*; *Kopec et al., 2015*; *Erlich et al., 2011*; *Gao et al., 2018*, |
| mPFC (PL/ILA) (medial prefrontal cortex) | *Liu et al., 2014*; *Schmitt et al., 2017*, |
| | *Bolkan et al., 2017* |
| OFC (orbitofrontal cortex) | *Wu et al., 2020* |
| PPC (VISa, VISam) (posterior parietal cortex) | *Voitov and Mrsic-Flogel, 2022*; *Harvey et al., 2012* |
| AIa (AId, AIv) | *Zhu et al., 2020* |
| Area p (VISpl) | *Gilad et al., 2018* |
| Dorsal cortex | *Pinto et al., 2019* |
| Entorhinal (in vitro persistent activity) | *Egorov et al., 2002* |
| Piriform | *Zhang et al., 2019*; *Wu et al., 2020* |
| VM/VAL (ventromedial nucleus, ventroanterior-lateral complex) | *Guo et al., 2017* |
| MD (mediodorsal nucleus) | *Schmitt et al., 2017*; *Bolkan et al., 2017* |
| Superior colliculus | *Kopec et al., 2015* |
| Cerebellar nucleus | *Gao et al., 2018* |

In this article, we report a cortical mechanism of distributed working memory that does not depend on a gradient of synaptic excitation. We developed an anatomically-based model of the mouse brain for working memory, built on the recently available mesoscopic connectivity data of the mouse thalamocortical system (*Oh et al., 2014*; *Gămănut et al., 2018*; *Harris et al., 2019*; *Kim et al., 2017*). Our model is validated by capturing large-scale neural activity observed in recent mouse experiments (*Guo et al., 2017*; *Gilad et al., 2018*). Using this model, we found that a decreasing gradient of synaptic inhibition mediated by parvalbumin (PV)-positive GABAergic cells (*Kim et al., 2017*; *Fulcher et al., 2019*; *Wang, 2020*) and long-range excitatory connections shapes the distributed pattern of working memory representation. Moreover, the engagement of inhibition through local and long-range projections determines the stability of the local circuits, further emphasizing the importance of inhibitory circuits.

A focus of this work is to examine whether anatomical connectivity can predict the emergent large-scale neural activity patterns underlying working memory. Interestingly, traditional graph-theory measures of interareal connections, which ignore cell types of projection targets, are uncorrelated with activity patterns. We propose new cell type-specific graph theory measures to overcome this problem and differentiate contributions of cortical areas in terms of their distinct roles in loading, maintaining, and reading out the content of working memory. Through computer-simulated perturbations akin to optogenetic inactivations, a core subnetwork was uncovered for the generation of persistent activity. This core subnetwork can be predicted based on the cell type-specific interareal connectivity, highlighting the necessity of knowing the cell type targets of interareal connections in order to relate anatomy with physiology and behavior. This work provides a computational and theoretical platform for cross-scale understanding of cognitive processes across the mouse cortex.

## Results

### A decreasing gradient of PV interneuron density from sensory to association cortex

Our large-scale circuit model of the mouse cortex uses interareal connectivity provided by anatomical data within the 43-area parcellation in the common coordinate framework v3 atlas (*Oh et al., 2014*; *Figure 1A*, *Figure 1—figure supplement 1A*). The model is endowed with area-to-area variation of PV-expressing interneurons in the form of a gradient measured from the qBrain mapping platform (*Figure 1—figure supplement 1B*; *Kim et al., 2017*). The PV cell density (the number of PV cells per unit volume) is divided by the total neuron density to give the PV cell fraction, which better reflects the expected amount of synaptic inhibition mediated by PV neurons (*Figure 1B and C*, neuron density is shown in *Figure 1—figure supplement 1C*). Cortical areas display a hierarchy defined by mesoscopic connectome data acquired using anterograde fluorescent tracers (*Oh et al., 2014*; *Figure 1D and E*). In *Figure 1F*, the PV cell fraction is plotted as a function of the cortical hierarchy, which shows a moderate negative correlation between the two. Therefore, primary sensory areas have a higher density of PV interneurons than association areas, although the gradient of PV interneurons does not align perfectly with the cortical hierarchy.

### A whole-mouse cortex model with a gradient of interneurons

In our model, each cortical area is described by a local circuit (*Figure 2A*) using a mean-field reduction (*Wong and Wang, 2006*) of a spiking neural network (*Wang, 2002*). Specifically, each cortical area contains two excitatory neural pools selective for different stimuli and a shared inhibitory neural pool. The model makes the following assumptions. First, local inhibitory strength is proportional to PV interneuron density across the cortex. Second, the interareal long-range connection matrix is given by the anterograde tracing data (*Oh et al., 2014*; *Knox et al., 2019*; *Wang et al., 2020*). Third, targeting is biased onto inhibitory cells for top-down compared with bottom-up projections. Therefore, feedforward connections have a greater net excitatory effect than feedback connections, which is referred to as the counterstream inhibitory bias (CIB) (*Mejías and Wang, 2022*; *Javadzadeh and Hofer, 2022*; *Wang, 2022*). Briefly, we assume that long-range connections are scaled by a coefficient that is based on the hierarchy of the source and target areas. According to the CIB assumption, long-range connections to inhibitory neurons are stronger for feedback connections and weaker for feedforward connections, while the opposite holds for long-range connections to excitatory neurons.

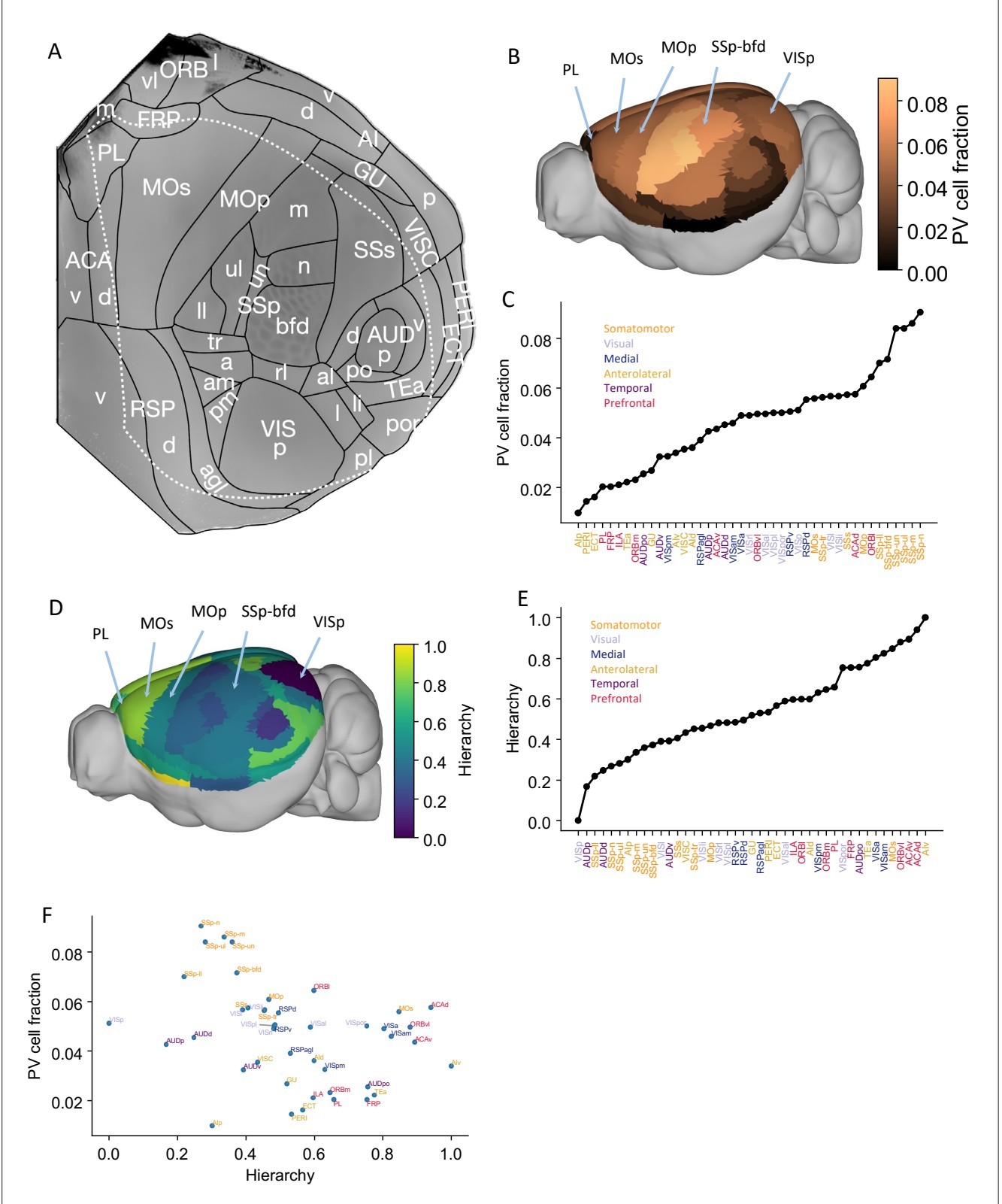

**Figure 1.** Anatomical basis of the multiregional mouse cortical model. (**A**) Flattened view of mouse cortical areas. Figure adapted from *Harris et al., 2019*. (**B**) Normalized parvalbumin (PV) cell fraction for each brain area, visualized on a 3D surface of the mouse brain. Five areas are highlighted: VISp, primary somatosensory area, barrel field (SSp-bfd), primary motor (MOp), MOs, and PL. (**C**) The PV cell fraction for each cortical area, ordered. Each area belongs to one of five modules, shown in color (*Harris et al., 2019*). (**D**) Hierarchical position for each area on a 3D brain surface. Five areas are

*Figure 1 continued on next page*

*Figure 1 continued*

highlighted as in (**B**), and color represents the hierarchy position. (**E**) Hierarchical positions for each cortical area. The hierarchical position is normalized and the hierarchical position of VISp is set to be 0. As in (**C**), the colors represent the module that an area belongs to. (**F**) Correlation between PV cell fraction and hierarchy (Pearson correlation coefficient $r = -0.35$, p<0.05).

The online version of this article includes the following figure supplement(s) for figure 1:

**Figure supplement 1.** Anatomical details of the mouse cortex.

## Distributed working memory activity depends on the gradient of inhibitory neurons and the cortical hierarchy

We simulated the large-scale network to perform a simple visual delayed response task that requires one of two stimuli to be held in working memory. We shall first consider the case in which the strength of local recurrent excitation is insufficient to generate persistent activity when parcellated areas are disconnected from each other. Consequently, the observed distributed mnemonic representation must depend on long-range interareal excitatory connection loops. Later in the article, we will discuss the network model behavior when some local areas are capable of sustained persistent firing in isolation.

The main question is: when distributed persistent activity emerges after a transient visual input is presented to the primary visual cortex (VISp), what determines the spatial pattern of working memory representation? After removal of the external stimulus, the firing rate in area VISp decreases rapidly to baseline. Neural activity propagates throughout the cortex after stimulus offset (*Figure 2B*). Neural activities in the higher visual cortical areas (e.g., VISrl and VISpl) show similar dynamics to VISp. In stark contrast, many frontal and lateral areas (including prelimbic [PL], infralimbic [ILA], secondary motor [MOs], and ventral agranular insula [AIv] areas) sustained a high firing rate during the delay period (*Figure 2B*). Areas that are higher in the cortical hierarchy show elevated activity during the delay period (*Figure 2C*). This persistent firing rate could last for more than 10 s and is a stable attractor state of the network (*Inagaki et al., 2019*).

The cortical hierarchy and PV fraction predict the delay-period firing rate of each cortical area (*Figure 2C–E*). Thus the activity pattern of distributed working memory depends on both local and large-scale anatomy. The delay activity pattern has a stronger correlation with hierarchy ($r = 0.91$) than with the PV fraction ($r = -0.43$). The long-range connections thus play a predominant important role in defining the persistent activity pattern.

Activity in early sensory areas such as VISp displays a rigorous response to the transient input but returns to a low firing state after stimulus withdrawal. In contrast, many frontal areas show strong persistent activity. When the delay-period firing rates are plotted versus hierarchy, we observe a gap in the distribution of persistent activity (*Figure 2D*) that marks an abrupt transition in the cortical space. This leads to the emergence of a subnetwork of areas capable of working memory representations.

We also used our circuit model to simulate delayed response tasks with different sensory modalities (*Figure 2—figure supplement 1*) by stimulating primary somatosensory area SSp-bfd and primary auditory area AUDp. The pattern of delay-period firing rates for these sensory modalities is similar to the results obtained for the visual task: sensory areas show transient activity, while frontal and lateral areas show persistent activity after stimulus withdrawal. Moreover, the cortical hierarchy could predict the delay-period firing rate of each cortical area well ($r = 0.89$, p<0.05), while the PV cell fraction could also predict the delay-period firing rate of each cortical area with a smaller correlation coefficient ($r = -0.4$, p<0.05). Our model thus predicts that working memory may share common activation patterns across sensory modalities, which is partially supported by cortical recordings during a memory-guided response task (*Inagaki et al., 2018*).

We explored the potential contributions of PV gradients and CIB in determining spatially patterned activity across the cortex. To evaluate the importance of the PV gradient, we replaced the PV gradient across areas with a constant value (*Figure 3A(ii)*). As compared to the model with a PV gradient (*Figure 3A(i)*), we found that, during the delay period, the number of cortical areas displaying persistent activity is diminished, but the abrupt transition in delay-period firing rates remains. This quantitative difference depends on the constant value used to scale inhibition from PV cells across areas (*Figure 3—figure supplement 1A and B*). Next, we performed the analogous manipulation

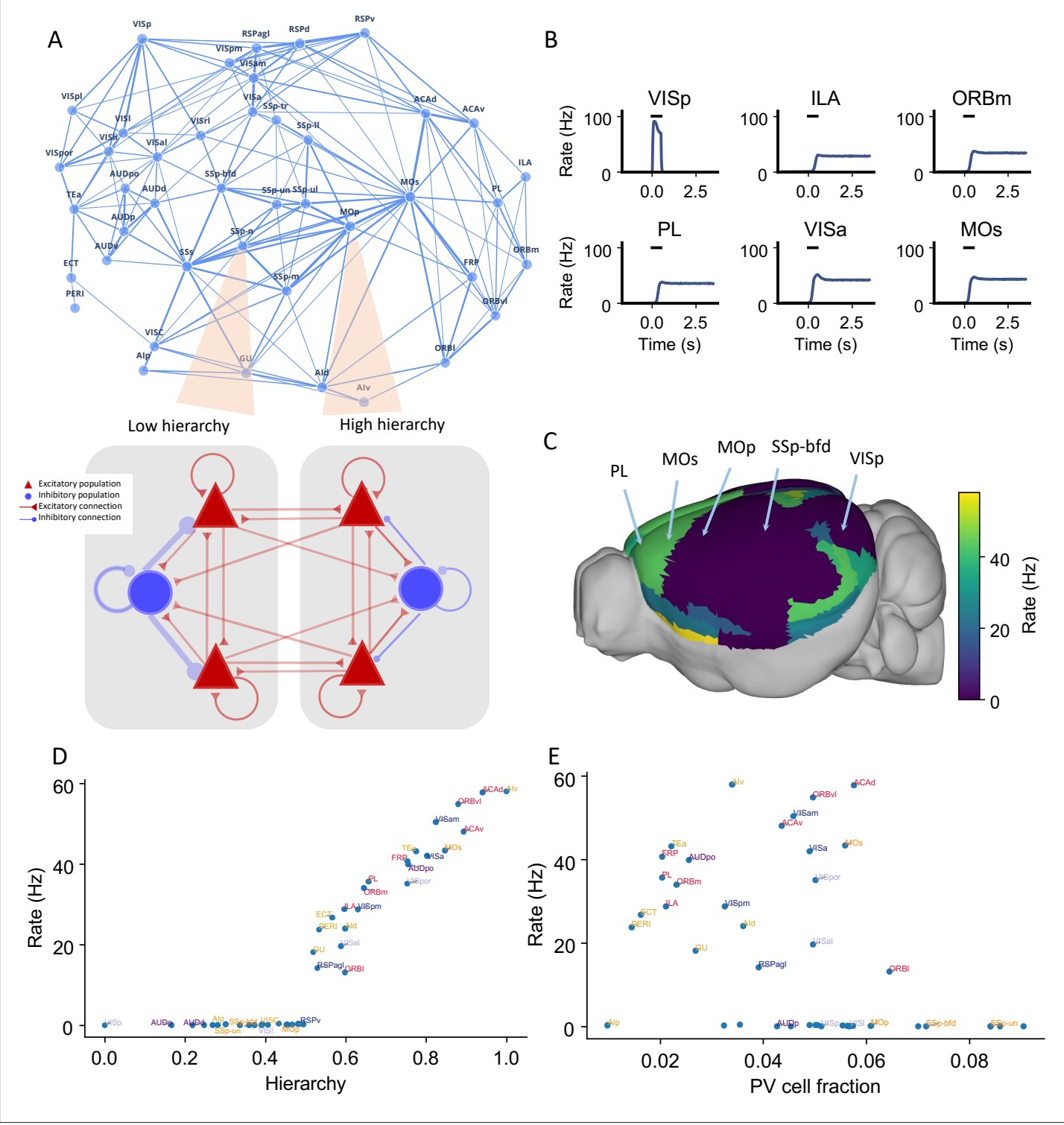

**Figure 2.** Distributed working memory activity depends on the gradient of parvalbumin (PV) interneurons and the cortical hierarchy. (**A**) Model design of the large-scale model for distributed working memory. Top: connectivity map of the cortical network. Each node corresponds to a cortical area and an edge is a connection, where the thickness of the edge represents the strength of the connection. Only strong connections are shown (without directionality for the sake of clarity). Bottom: local and long-range circuit design. Each local circuit contains two excitatory populations (red), each selective to a particular stimulus and one inhibitory population (blue). Long-range connections are scaled by mesoscopic connectivity strength (*Oh et al., 2014*) and follows counterstream inhibitory bias (CIB) (*Mejías and Wang, 2022*). (**B**) The activity of six selected areas during a working memory task is shown. A visual input of 500 ms is applied to area VISp, which propagates to the rest of the large-scale network. (**C**) Delay-period firing rate for

*Figure 2 continued on next page*

*Figure 2 continued*

each area on a 3D brain surface. Similar to *Figure 1B*, the positions of five areas are labeled. (**D**) Delay-period firing rate is positively correlated with cortical hierarchy (*r* = 0.91, p<0.05). (**E**) Delay-period firing rate is negatively correlated with PV cell fraction (*r* = –0.43, p<0.05).

The online version of this article includes the following figure supplement(s) for figure 2:

**Figure supplement 1.** Example simulation for different sensory modalities.

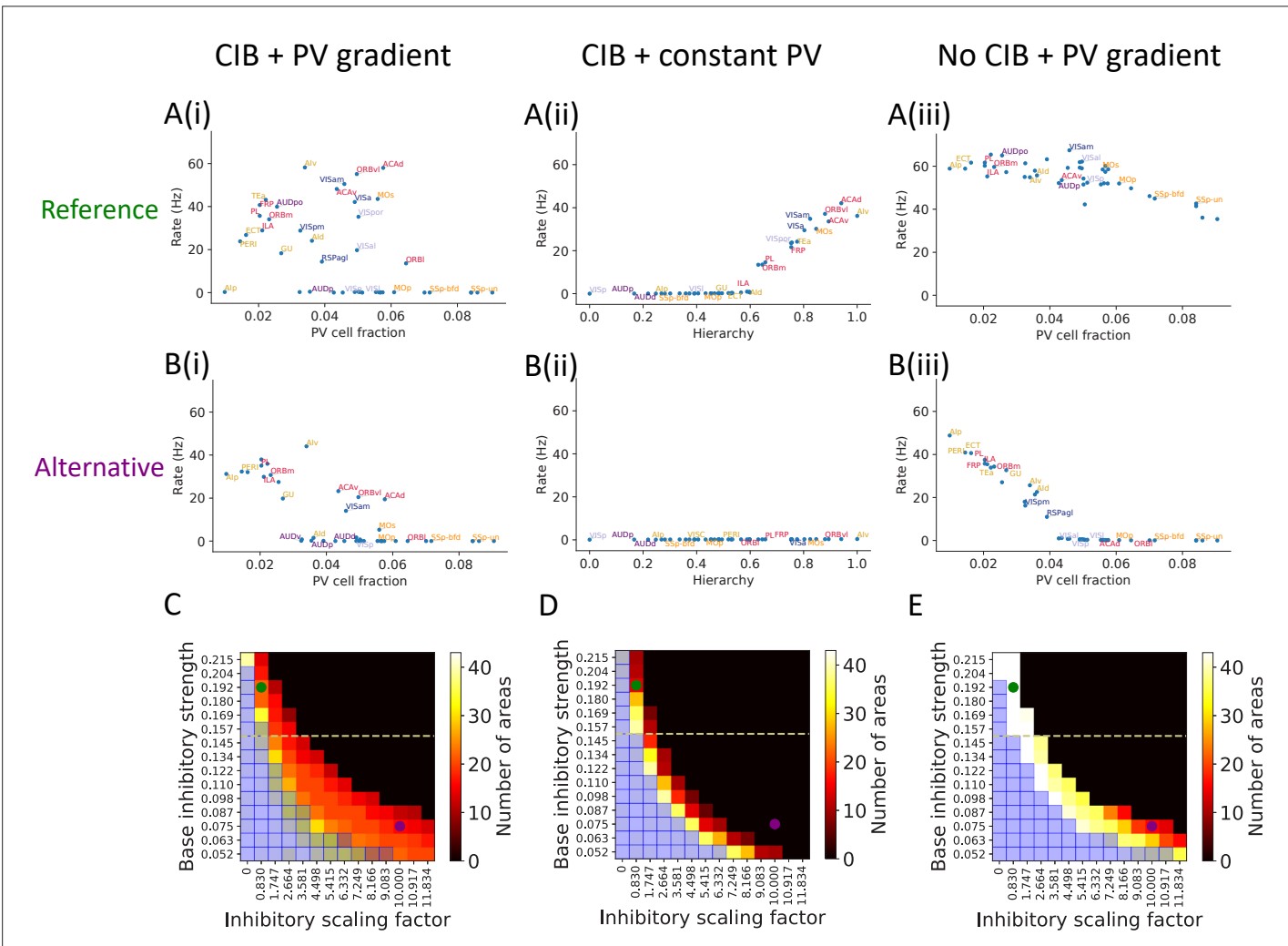

**Figure 3.** The role of parvalbumin (PV) inhibitory gradient and hierarchy-based counter inhibitory bias (CIB) in determining persistent activity patterns in the cortical network. (**A(i)**) Delay firing rate as a function of PV cell fraction with both CIB and PV gradient present (*r* = –0.42, p<0.05). This figure panel is the same as *Figure 2E*. (**A(ii)**) Delay firing rate as a function of hierarchy after removal of PV gradient (*r* = –0.85, p<0.05). (**A(iii)**) Delay firing rate as a function of PV cell fraction after removal of CIB (*r* = –0.74, p<0.05). (**B(i)**) Delay firing rate as a function of PV cell fraction with both CIB and PV gradient present, in the alternative regime (*r* = –0.7, p<0.05). (**B(ii)**) Delay firing rate as a function of hierarchy after removal of PV gradient, in the alternative regime (*r* = –0.95, p<0.05). (**B(iii)**) Delay firing rate as a function of PV cell fraction after removal of CIB, in the alternative regime (*r* = –0.84, p<0.05). (**C–E**) Number of areas showing persistent activity (color coded) as a function of the local inhibitory gradient ($g_{EI,scaling}$, X-axis) and the base value of the local inhibitory gradient ($g_{EI,0}$, Y-axis) for the following scenarios: (**C**) CIB and PV gradient, (**D**) with PV gradient replaced by a constant value, and (**E**) with CIB replaced by a constant value. The reference regime is located at the top-left corner of the heatmap (green dot) and corresponds to **A(i)–A(iii)**, while the alternative regime is located at the lower-right corner (purple dot) and corresponds to **B(i)–B(iii)**. The yellow dashed lines separate parameters sets for which none of the areas show 'independent' persistent activity (above the line) from parameter sets for which some areas are capable of maintaining persistent activity without input from other areas (below the line). Blue shaded squares in the heatmap mark the absence of a stable baseline.

The online version of this article includes the following figure supplement(s) for figure 3:

**Figure supplement 1.** Dependence of persistent activity on inhibitory model parameters.

on the CIB by scaling feedforward and feedback projections with a constant value across areas, thus effectively removing the CIB. In this case, the firing rate of both sensory and association areas exhibit high firing rates during the delay period (***Figure 3A(iii)***). Thus, CIB may be particularly important in determining which areas exhibit persistent activity.

To further explore the model parameter space and better understand the interplay between PV gradient and CIB, we systematically varied two critical model parameters: (i) the base local inhibitory weight $g_{EI,0}$ onto excitatory neurons, which sets the minimal inhibition for each cortical area, and (ii) the scaling factor $g_{EI,scaling}$, which refers to how strongly the PV gradient is reflected in the inhibitory weights. We created heatmaps that show the number of areas with persistent activity during the delay period as a function of these parameters: in ***Figure 3C***, we simulate the network with both CIB and PV gradient, while in ***Figure 3D and E*** we simulate networks when PV gradient or CIB is removed, respectively. In each of these networks, we identify two regimes based on specific values for $g_{EI,0}$ and $g_{EI,scaling}$: a reference regime (used throughout the rest of the article) and an alternative regime.

If we remove the PV gradient in the alternative parameter regime, persistent activity is lost (***Figure 3B(ii)***). In contrast, if we remove CIB the model still exhibits an abrupt transition in firing rate activity (***Figure 3B(iii)***). In this regime, a strong correlation and piece-wise linear relationship between firing rate and PV cell fraction was uncovered that did not exist when CIB was present. This observation led to a model prediction: if PV cell fraction is not strongly correlated with delay firing rate across cortical areas (e.g., ***Figure 3A(i)*** or ***Figure 3B(i)***), this suggests the existence of a CIB mechanism at play. Importantly, the model without CIB exhibits the abrupt transition in delay-period firing rates provided it is in a regime where some areas exhibit 'independent' persistent activity: persistent activity that is generated due to local recurrence and thus independent of long-range recurrent loops. The parameter regime where some areas exhibit 'independent' persistent activity is quantified by varying the base value of local inhibitory connections (***Figure 3—figure supplement 1C***). To conclude, the model results suggest that CIB may be present in a large-scale brain network if the PV cell fraction is not strongly correlated with the delay firing rate. Furthermore, CIB may be particularly important in the regime where local connections are not sufficient to sustain independent persistent activity.

Next, we evaluated the stability of the baseline state for the three conditions described above: (i) original with PV gradient and CIB, (ii) after removal of CIB, and (iii) after removal of PV gradient. The heatmaps obtained after varying the base inhibitory strength and inhibitory scaling factor were qualitatively the same across the three conditions, as shown by the blue shaded squares in ***Figure 3C–E***. There are some regimes, such as the one depicted on the lower-left corner, where all areas exhibit persistent activity and there is no stable baseline: a regime that is not biologically realistic for a healthy brain. Thus, while PV and CIB shape the distribution of delay firing rates across cortical areas, they do not qualitatively influence the system's baseline stability. However, the inclusion of both PV gradient and CIB in the model (***Figure 3C***) results in a more robust system, that is, a far wider set of parameters can produce realistic persistent activity (***Figure 3C, D and E***).

## Local and long-range projections modulate the stability of the baseline state in the cortex

The stability of the baseline state for any given cortical area may have contributions from local inhibition or from long-range projections that target local inhibitory circuits. We found that individual local networks without long-range connections are stable without local inhibition (***Figure 4A***, see 'Methods' for theoretical calculation of stability in a local circuit). However, in the full network with long-range connections, setting either the long-range connections to inhibitory neurons or local inhibition to zero made the network's baseline state unstable, and individual areas rose to a high firing rate (***Figure 4B***). Thus, inhibition from local and long-range circuits contributes to the baseline stability of cortical areas.

Motivated by the results on stability, we investigated whether the large-scale network model operates in the inhibitory stabilized network (ISN) regime (***Tsodyks et al., 1997***; ***Sanzeni et al., 2020***), whereby recurrent excitation is balanced by inhibition to maintain stability of the baseline state. First, we examined whether individual brain areas (i.e., without long-range projections) may operate in this regime. We found a parameter set in which the baseline firing rate is stable only when local inhibition is intact: when inhibition is removed, the stable baseline state disappears, which suggests that the local circuits are ISNs (***Figure 4C*** and see stability analysis in the 'Methods' section). In the full neural network with long-range connections, similar analysis as in ***Figure 4B*** shows that the network

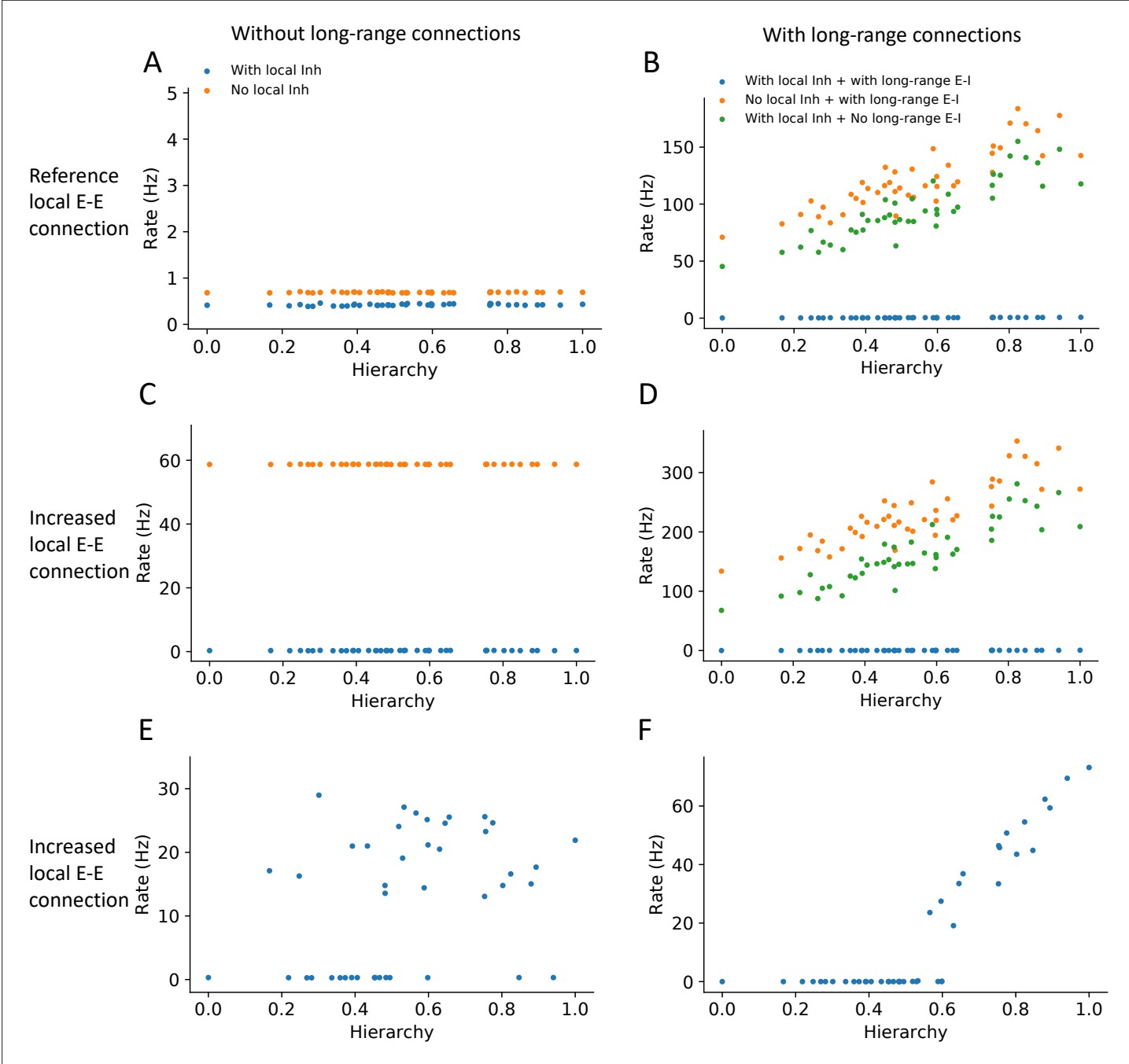

**Figure 4.** Local and long-range projections modulate the baseline stability of individual cortical areas. Steady state firing rates are shown as a function of hierarchy for different scenarios: (**A**) without long-range connections in the reference regime ($g_{E,self} = 0.4nA$, $g_{EI,0} = 0.192nA$), (**B**) with long-range connections in the reference regime ($\mu_{EE} = 0.1nA$), (**C**) without long-range connections and increased local excitatory connections ($g_{E,self} = 0.6nA$, $g_{EI,0} = 0.5nA$), and (**D**) with long-range connections (increased long-range connections to excitatory neurons, $\mu_{EE} = 0.19nA$) and increased local excitatory connections. (**E**) Firing rate as a function of hierarchy when external input given to each area, showing bistability for a subset of areas (parameters as in (**C**) and 'with local inh'). (**F**) Firing rate as a function of hierarchy when external input is applied to area VISp (parameters as in (**D**) and 'with local inh + with long-range E-I').

becomes unstable if long-range projections onto inhibitory interneurons are removed (**Figure 4D**). Thus we propose that the network is also in a 'global' ISN regime, whereby long-range connections to inhibitory neurons are necessary to maintain a stable baseline state. Second, we examined whether the ISN regime is consistent with distributed working memory patterns in the cortex (**Figure 2**). In the

regime with increased local excitatory connections but without long-range projections, some local circuits could reach a high stable state when an external input is applied, demonstrating the bistability of those areas (*Figure 4E*). When we considered the full network with long-range projections, the network exhibits a graded firing rate pattern after transient stimulation of VISp, showing that the interconnected ISN networks are compatible with bistability of a subset of cortical areas (*Figure 4F*).

In summary, we have shown that distinct local and long-range inhibitory mechanisms shape the pattern of working memory activity and stability of the baseline state.

## Thalamocortical interactions maintain distributed persistent activity

To investigate how thalamocortical interactions affect the large-scale network dynamics, we designed a thalamocortical network similar to the cortical network (*Figure 5A*). Several studies have shown that thalamic areas are also involved in the maintenance of working memory (*Bolkan et al., 2017*; *Guo et al., 2017*; *Schmitt et al., 2017*). However, the large-scale thalamocortical mechanisms underlying memory maintenance are unknown. We set the strength of connections between the thalamus and cortex using data from the Allen Institute (*Oh et al., 2014*; *Figure 5—figure supplement 1*). All thalamocortical connections in the model are mediated by AMPA synapses. There are no recurrent connections in the thalamus within or across thalamic nuclei (*Jones, 2007*). The effect of thalamic reticular nucleus neurons was included indirectly as a constant inhibitory current to all thalamic areas (*Crabtree, 2018*; *Hádinger et al., 2023*). Similarly to cortical areas, the thalamus is organized along a measured hierarchy (*Harris et al., 2019*). For example, the dorsal part of the lateral geniculate nucleus (LGd) is lower than the cortical area VISp in the hierarchy, consistent with the fact that LGd sends feedforward inputs to VISp. Thalamocortical projections in the model are slightly more biased toward excitatory neurons in the target area if they are feedforward projections and toward inhibitory neurons if they are feedback.

Here, we weakened the strength of cortical interareal connections as compared to the cortex model of *Figure 2*. Now, persistent activity can still be generated (*Figure 5B*, blue) but is maintained with the help of the thalamocortical loop, as observed experimentally (*Guo et al., 2017*). Indeed, in simulations where the thalamus was inactivated, the cortical network no longer showed sustained activity (*Figure 5B*, red).

In the thalamocortical model, the delay activity pattern of the cortical areas correlates with the hierarchy, again with a gap in the firing rate separating the areas engaged in persistent activity from those that do not (*Figure 5B and C*). Sensory areas show a low delay firing rate, and frontal areas show strong persistent firing. Unlike the cortex, the firing rate of thalamic areas continuously increases along the hierarchy (*Figure 5E*). On the other hand, cortical dynamics in the thalamocortical and cortical models show many similarities. Early sensory areas do not show persistent activity in either model. Many frontal and lateral areas show persistent activity and there is an abrupt transition in cortical space in the thalamocortical model, like in the cortex-only model. Quantitatively, the delay firing pattern of the cortical areas is correlated with the hierarchy and the PV fraction (*Figure 5C and D*). Furthermore, the delay-period firing rate of cortical areas in the thalamocortical model correlates well with the firing rate of the same areas in the cortical model (*Figure 5F*). This comparison suggests that the cortical model captures most of the dynamical properties in the thalamocortical model; therefore in the following analyses, we will mainly focus on the cortex-only model for simplicity.

## Cell type-specific connectivity measures predict distributed persistent firing patterns

Structural connectivity constrains large-scale dynamics (*Mejías and Wang, 2022*; *Froudist-Walsh et al., 2021*; *Cabral et al., 2011*). However, we found that standard graph theory measures could not predict the pattern of delay-period firing across areas. There is no significant correlation between input strength (the total weight of all interareal input connections to an area) and delay-period firing rate ($r = 0.25$, p=0.25, *Figure 6A(i) and (ii)*) and input strength cannot predict which areas show persistent activity (prediction accuracy = 0.51, *Figure 6A(iii)*). We hypothesized that this is because currently available connectomic data used in this model do not specify the type of neurons targeted by the long-range connections. For instance, when two areas are strongly connected with each other, such a loop would contribute to the maintenance of persistent activity if projections are mutually excitatory, but not if one of the two projections predominantly targets inhibitory PV cells. Therefore,

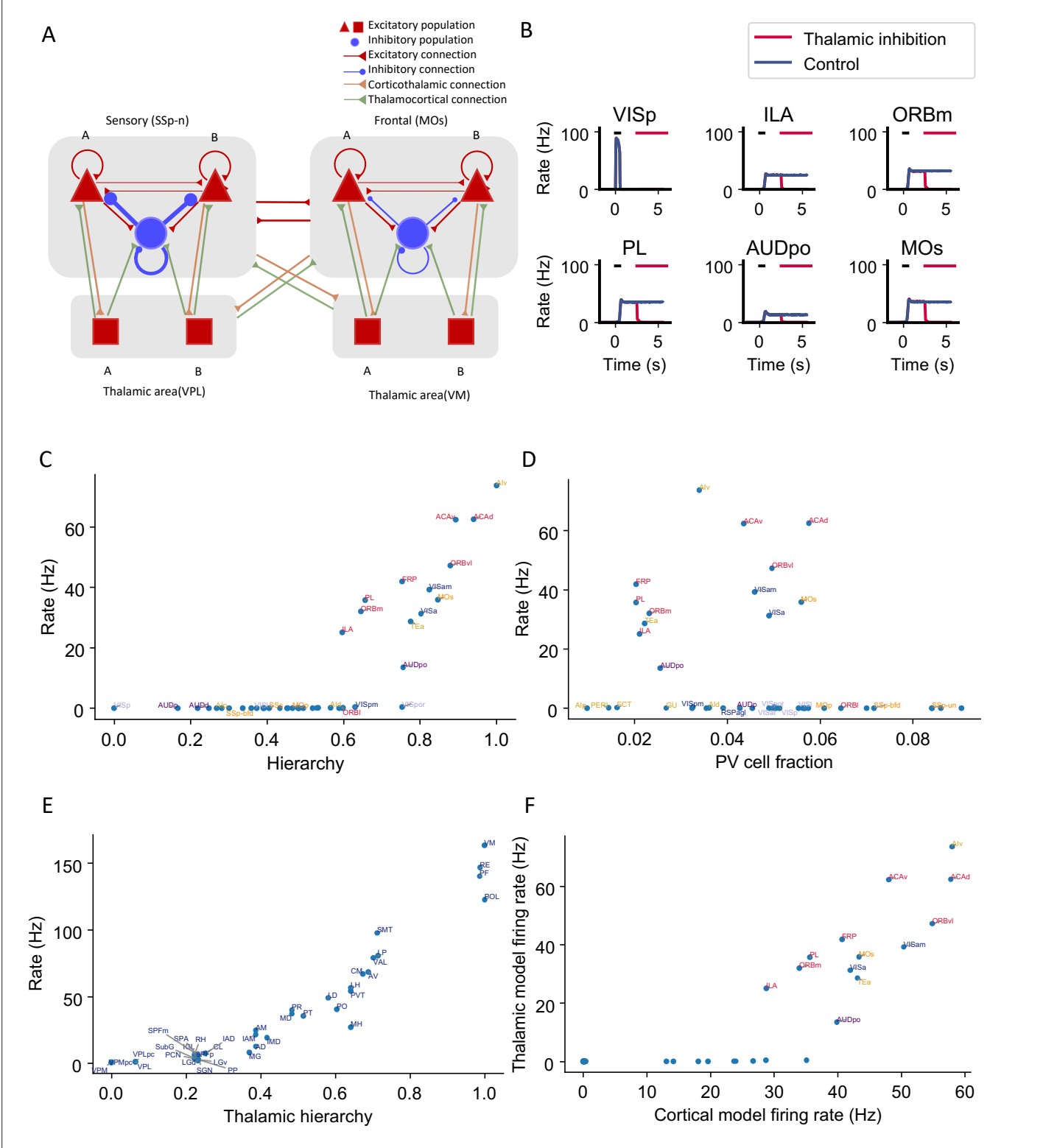

**Figure 5.** Thalamocortical interactions help maintain distributed persistent activity. (**A**) Model schematic of the thalamocortical network. The structure of the cortical component is the same as our default model in *Figure 2A*, but with modified parameters. Each thalamic area includes two excitatory populations (red square) selective to different stimuli. Long-range projections between thalamus and cortex also follow the counterstream inhibitory bias rule as in the cortex. Feedforward projections target excitatory neurons with stronger connections and inhibitory neurons with weaker connections; the opposite holds for feedback projections. (**B**) The activity of six sample cortical areas in a working memory task is shown during control (blue) and

*Figure 5 continued on next page*

*Figure 5 continued*

when thalamic areas are inhibited in the delay period (red). Black dashes represent the external stimulus applied to VISp. Red dashes represent external inhibitory input given to all thalamic areas. (**C**) Delay-period firing rate of cortical areas in the thalamocortical network. The activity pattern has a positive correlation with cortical hierarchy ($r = 0.78$, p<0.05). (**D**) Same as (**C**) but plotted against parvalbumin (PV) cell fraction. The activity pattern has a negative correlation with PV cell fraction, but it is not significant ($r = –0.26$, p=0.09). (**E**) Delay firing rate of thalamic areas in thalamocortical network. The firing rate has a positive correlation with thalamic hierarchy ($r = 0.94$, p<0.05). (**F**) Delay-period firing rate of cortical areas in thalamocortical network has a positive correlation with delay firing rate of the same areas in a cortex-only model ($r = 0.77$, p<0.05). Note that only the areas showing persistent activity in both models are considered for correlation analyses.

The online version of this article includes the following figure supplement(s) for figure 5:

**Figure supplement 1.** Anatomical data of thalamus and cortical connectivity.

cell type specificity of interareal connections must be taken into account in order to relate the connectome with the whole-brain dynamics and function. To examine this possibility, we introduced a *cell type projection coefficient* (see 'Calculation of network structure measures' in the 'Methods' section), which is smaller when the target area has a higher PV cell fraction (*Figure 6—figure supplement 1*). The cell type projection coefficient also takes cell type targets of long-range connections into account, which, in our model, is quantified by CIB. As a result, the modified cell type-specific connectivity measures increase if the target area has a low density of PV interneurons and/or if long-range connections predominantly target excitatory neurons in the target area.

We found that cell type-specific graph measures accurately predict delay-period firing rates. The cell type-specific input strength of the early sensory areas is weaker than the raw input strength (*Figure 6B(i)*). Importantly, the firing rate across areas is positively correlated with cell type-specific input strength (*Figure 6B(ii)*). Cell type-specific input strength also accurately predicts which areas show persistent activity (*Figure 6B(iii)*). Similarly, we found that the cell type-specific eigenvector centrality, but not standard eigenvector centrality (*Newman, 2018*), was a good predictor of delay-period firing rates (*Figure 6—figure supplement 2*).

## A core subnetwork for persistent activity across the cortex

Many areas show persistent activity in our model. However, are all active areas equally important in maintaining persistent activity? When interpreting large-scale brain activity, we must distinguish different types of contribution to working memory. For instance, inactivation of an area like VISp impairs performance of a delay-dependent task because it is essential for a (visual) 'input' to access working memory; on the other hand, a 'readout' area may display persistent activity only as a result of sustained inputs from other areas that form a 'core,' which are causally important for maintaining a memory representation.

We propose four types of areas related to distributed working memory: input, core, readout, and nonessential (*Figure 7A*). External stimuli first reach input areas, which then propagate activity to the core and nonessential areas. Core areas form recurrent loops and support distributed persistent activity across the network. By definition, disrupting any of the core areas would affect persistent activity globally. The readout areas also show persistent activity. Yet, inhibiting readout areas has little effect on persistent activity elsewhere in the network. We can assign the areas to the four classes based on three properties: (i) the effect of inhibiting the area during stimulus presentation on delay activity in the rest of the network; (ii) the effect of inhibiting the area during the delay period on delay activity in the rest of the network; and (iii) the delay activity of the area itself on trials without inhibition.

In search of a core working memory subnetwork in the mouse cortex, in model simulations we inactivated each area either during stimulus presentation or during the delay period, akin to optogenetic inactivation in mice experiments. The effect of inactivation was quantified by calculating the decrement in the firing rate compared to control trials for the areas that were not inhibited (*Figure 7B*). The VISp showed a strong inhibition effect during the stimulus period, as expected for an input area. We identified seven areas with a substantial inhibition effect during the delay period (*Figure 7C*), which we identify as a core for working memory. Core areas are distributed across the cortex. They include frontal areas PL, ILA, medial part of the orbital area (ORBm), which are known to contribute to working memory (*Liu et al., 2014*; *Bolkan et al., 2017*). Other associative and sensory areas (AId, VISpm, ectorhinal area [ECT], gustatory area [GU]) are also in the core. Similarly, we used the above criteria to classify areas as readout or nonessential (*Figure 7D*).

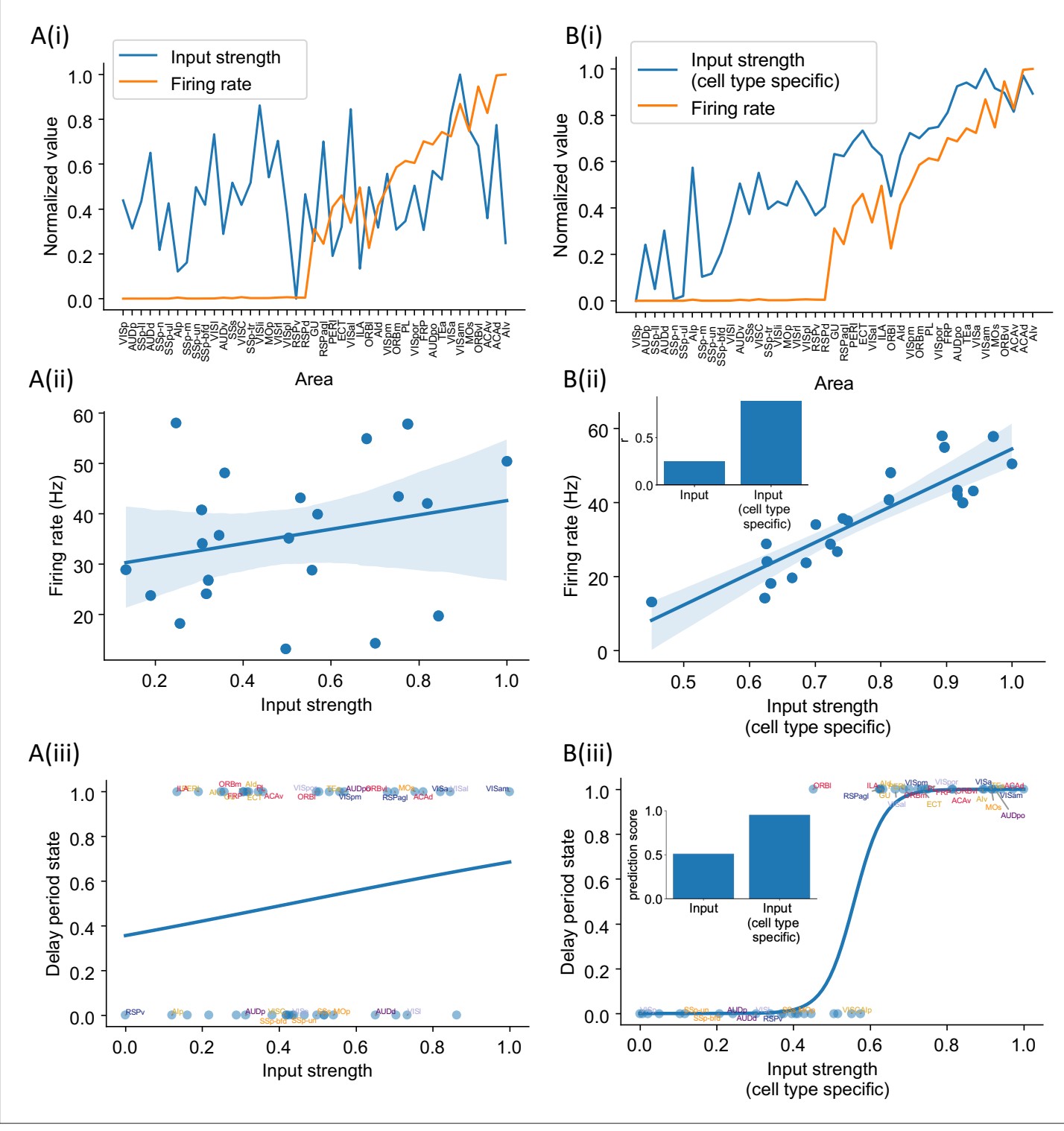

**Figure 6.** Cell type-specific connectivity measures are better at predicting firing rate pattern than nonspecific ones. (**A(i)**) Delay-period firing rate (orange) and input strength for each cortical area. Input strength of each area is the sum of connectivity weights of incoming projections. Areas are plotted as a function of their hierarchical positions. Delay-period firing rate and input strength are normalized for better comparison. (**A(ii)**) Input strength does not show significant correlation with delay-period firing rate for areas showing persistent activity in the model ($r = 0.25$, $p=0.25$). (**A(iii)**) Input strength cannot be used to predict whether an area shows persistent activity or not (prediction accuracy = 0.51). (**B(i)**) Delay-period firing rate (orange) and cell type-specific input strength for each cortical area. Cell type-specific input strength considers how the long-rang connections target different cell types and is the sum of modulated connectivity weights of incoming projections. Same as (**A(i)**), areas are sorted according to their

*Figure 6 continued on next page*

*Figure 6 continued*

hierarchy and delay-period firing rate and input strength are normalized for better comparison. (**B(ii)**) Cell type-specific input strength has a strong correlation with delay-period firing rate of cortical areas showing persistent activity ($r = 0.89$, $p<0.05$). Inset: comparison of the correlation coefficient for raw input strength and cell type-specific input strength. (**B(iii)**) Cell type-specific input strength predicts whether an area shows persistent activity or not (prediction accuracy = 0.95). Inset: comparison of the prediction accuracy for raw input strength and cell type-specific input strength.

The online version of this article includes the following figure supplement(s) for figure 6:

**Figure supplement 1.** Details of cell type-specific connectivity measures.

**Figure supplement 2.** Cell type-specific eigenvector centrality measures are better at predicting firing rate patterns than raw eigenvector centrality measures.

**Figure supplement 3.** Sign-only input strength measure and noPV input strength measure predict firing rate well.

We have defined a core area for working memory maintenance as a cortical area that, first, exhibits persistent activity, and second, removal of this area (e.g., experimentally via a lesion or opto-inhibition) significantly affects persistent activity in other areas. It is possible, however, that effects on persistent activity at the network level only arise after lesioning two or more areas. Thus, we proceeded with inhibiting two, three, and four readout areas concurrently (*Figure 7—figure supplement 1A*), as by definition, inhibiting any single readout area will not exhibit a strong inhibition effect.

We first inhibited pairs of readout areas and evaluated the effect of this manipulation at a network level. Specifically, for any given readout area A, we plotted the average firing rate of the network when A was inhibited as part of an inhibited pair (see description in 'Methods'). After inhibiting a pair of readout areas, there was a decrement in the average firing rate of the network (*Figure 7—figure supplement 1A*). The decrement became more pronounced as more readout areas were inhibited, for example, triplets and quadruplets, and when a combination of readout and core areas was inhibited pairwise (*Figure 7—figure supplement 1B*). This analysis demonstrates that readout areas also play a role in maintaining distributed persistent activity: we may define 'second-order core areas' as those readout areas that have a strong inhibition effect only when inhibited concurrently with another area, while third-order and fourth-order core areas are analogously defined via triplet and quadruplet inhibition, respectively. We note that the effects of silencing pairs, triplets and quadruplets, of readout areas remain smaller than those seen after silencing single core areas listed above. We also tested the effect of inhibiting all core areas during the delay period (*Figure 7—figure supplement 1C*). After inhibiting all core areas, some readout areas lost persistent firing. Moreover, there was a 48% decrement in the average firing rate compared with a 15% decrement for a single core area and a 3% decrement for a single readout area. Thus, the pattern of persistent activity is more sensitive to perturbations of core areas, which underscores the classification of some cortical areas into core vs readout.

## The core subnetwork can be identified by the presence of strong excitatory loops

Inhibition protocols across many areas are computationally costly. We sought a structural indicator that is easy to compute and is predictive of whether an area is engaged in working memory function. Such an indicator could also guide the interpretation of large-scale neural recordings in experimental studies. In the dynamical regime where individual cortical areas do not show persistent activity independently, distributed working memory patterns must be a result of long-range recurrent loops across areas. We thus introduced a quantitative measurement of the degree to which each area is involved in long-range recurrent loops (*Figure 8A*).

The core subnetwork can be identified by the presence of strong loops between excitatory cells. Here we focus on length 2 loops (*Figure 8A*); the strength of a loop is the product of two connection weights for a reciprocally connected pair of areas; and the loop strength measure of an area is the sum of the loop strengths of all length 2 loops that the area is part of. Results were similar for longer loops (*Figure 8B*, also see *Figure 8—figure supplement 1* for results of longer loops). The raw loop strength had no positive statistical relationship to the core working memory subnetwork (*Figure 8C(i) and (ii)*). We then defined cell type-specific loop strength (see 'Methods'). The cell type-specific loop strength is the loop strength calculated using connectivity multiplied by the cell type projection coefficient. The cell type-specific loop strength, but not the raw loop strength, predicts which area is a core area with high accuracy (*Figure 8D(i) and (ii)*, prediction accuracy = 0.93). This demonstrates

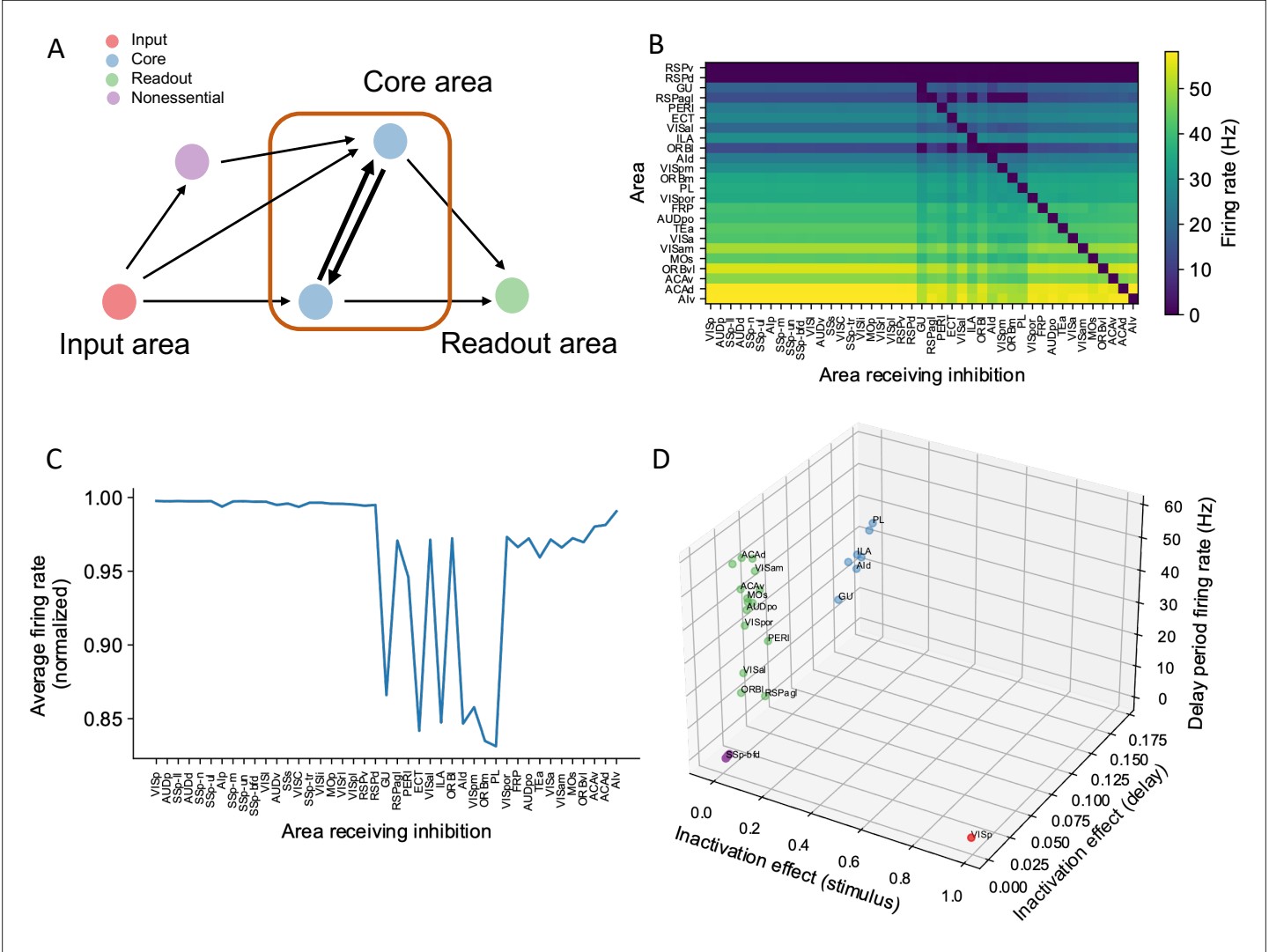

**Figure 7.** A core subnetwork generates persistent activity across the cortex. (**A**) We propose four different types of areas. Input areas (red) are responsible for coding and propagating external signals, which are then propagated through synaptic connections. Core areas (blue) form strong recurrent loops and generate persistent activity. Readout areas (green) inherit persistent activity from core areas. Nonessential areas (purple) may receive inputs and send outputs but they do not affect the generation of persistent activity. (**B**) Delay-period firing rate for cortical areas engaged in working memory (Y-axis) after inhibiting different cortical areas during the delay period (X-axis). Areas in the X-axis and Y-axis are both sorted according to hierarchy. Firing rates of areas with small firing rate (<1 Hz) are partially shown (only RSPv and RSPd are shown because their hierarchical positions are close to areas showing persistent activity). (**C**) The average firing rate for areas engaged in persistent activity under each inhibition simulation. The X-axis shows which area is inhibited during the delay period, and the Y-axis shows the average delay period activity for all areas showing persistent activity. Note that when calculating the average firing rate, the inactivated area was excluded in order to focus on the inhibition effect of one area on other areas. Average firing rates on the Y-axis are normalized using the average firing in control (no inhibition) simulation. (**D**) Classification of four types of areas based on their delay period activity after stimulus- and delay-period inhibition (color denotes the type for area, as in **A**). The inhibition effect, due to either stimulus or delay period inhibition, is the change of average firing rate normalized by the average firing rate in the control condition. Areas with strong inhibition effect during stimulus period are classified as input areas; areas with strong inhibition effect during delay period and strong delay-period firing rate are classified as core areas; areas with weak inhibition effect during delay period but strong delay-period firing rate during control are classified as readout areas; and areas with weak inhibition effect during delay period and weak delay-period firing rate during control are classified as nonessential areas.

The online version of this article includes the following figure supplement(s) for figure 7:

**Figure supplement 1.** Multiple-area inhibition experiments demonstrate the relative importance for core and readout areas in maintaining network-level persistent activity.

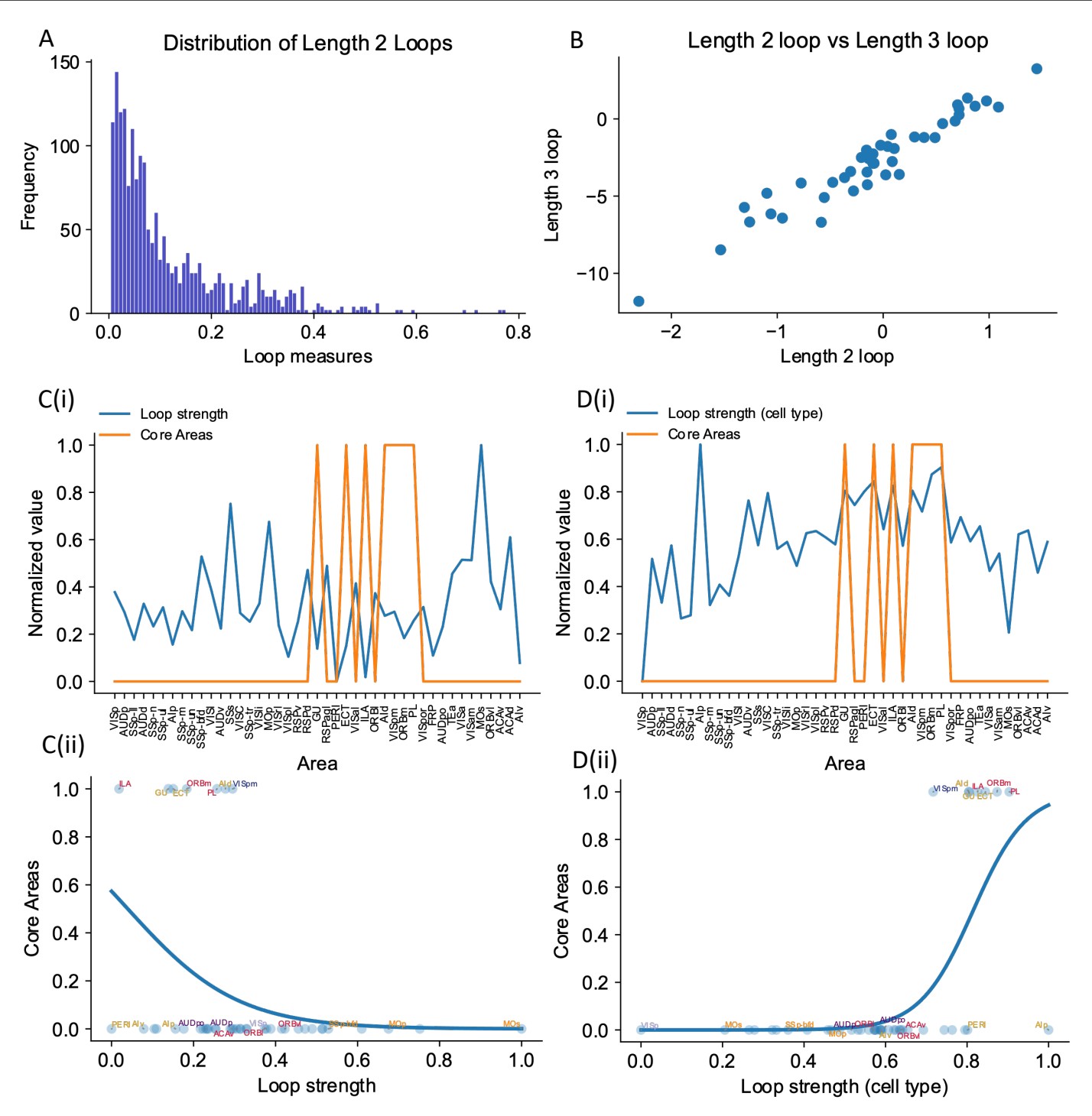

**Figure 8.** The core subnetwork can be identified structurally by the presence of strong excitatory loops. (**A**) Distribution of length 2 loops. X-axis is the single-loop strength of each loop (product of connectivity strengths within loop) and Y-axis is their relative frequency. (**B**) Loop strengths of each area calculated using different length of loops (e.g., length 3 vs length 2) are highly correlated ($r = 0.96$, $p<0.05$). (**C(i)**) Loop strength (blue) is plotted alongside core areas (orange), a binary variable that takes the value 1 if the area is a core area, 0 otherwise. Areas are sorted according to their hierarchy. The loop strength is normalized to a range of (0, 1) for better comparison. (**C(ii)**) A high loop strength value does not imply that an area is a core area. Blue curve shows the logistic regression curve fits to differentiate the core areas versus non-core areas. (**D(i)**) Same as (**C**), but for cell type-specific loop strength. (**D(ii)**) A high cell type-specific loop strength predicts that an area is a core area (prediction accuracy = 0.93). Same as (**C**), but for cell type-specific loop strength.

*Figure 8 continued on next page*

*Figure 8 continued*

The online version of this article includes the following figure supplement(s) for figure 8:

**Figure supplement 1.** Cell type-specific loop strengths (length 3 loops) are also better at predicting firing rate patterns than raw loop measures.

**Figure supplement 2.** Sign-only loop strengths or noPV loop strengths are not performing well at predicting firing rate patterns.

that traditional connectivity measures are informative but not sufficient to explain dynamics during cognition in the mouse brain. Cell type-specific connectivity, and new metrics that account for such connectivity, are necessary to infer the role of brain areas in supporting large-scale brain dynamics during cognition.

To better demonstrate our cell type-specific connectivity measures, we have implemented two other measures for comparison: (a) a loop-strength measure that adds a 'sign' without further modification, and (b) a loop strength measure that takes hierarchical information – and not PV information – into account. These two graph-theoretic measures can be used to predict delay firing rate during a sensory working memory task, thus highlighting the importance of hierarchical information, which distinguishes excitatory from inhibitory feedback (*Figure 6—figure supplement 3*). On the other hand, the prediction of the core areas greatly depends on cell-type specificity: the sign-only and 'noPV' mechanisms do not reliably predict whether an area is a core area or not, especially in the case of calculating with length 3 loops, demonstrating the importance of cell type-specific connectivity measures (*Figure 8—figure supplement 2*).

## Multiple attractor states emerge from the mouse mesoscopic connectome and local recurrent interactions

Different tasks lead to dissociable patterns of internally sustained activity across the brain, described dynamically as distinct attractor states. Generally, attractor states may enable computations such as decision making and working memory (*Wang, 1999*; *Wang, 2002*; *Mejías et al., 2016*). Specifically, a given task may be characterized by a specific attractor landscape and thereby define different core areas for working memory, as introduced above. We developed a protocol to identify multiple attractor states, then analyzed the relationship between network properties and the attractor states (*Figure 9A–C*). For different parameters, the number of attractors and the spatial patterns defined by these attractors change. Two parameters are especially relevant here. These are the long-range connection strength ($\mu_{EE}$) and local excitatory connection strength ($g_{E,self}$). These parameters affect the number of attractors in a model of the macaque cortex (*Mejías and Wang, 2022*). Increasing the long-range connection strength decreases the number of attractors (*Figure 9D*). Stronger long-range connections implies that the coupling between areas is stronger. If areas are coupled with each other, the activity state of an area will be highly correlated to that of its neighbors. This leads to less variability and fewer attractors.

To quantify how the patterns of attractors change for different parameters, two quantities are introduced. The *attractor fraction* is the fraction of all detected attractor states to which an area belongs. An area 'belongs' to an attractor state if it is in a high activity state in that attractor. The *attractor size* is defined by the number of areas belonging to that attractor. As we increased the long-range connection strength, the attractor size distribution became bimodal. The first mode corresponded to large attractors, with many areas. The second mode corresponded to small attractors, with few areas (*Figure 9D*).

When the local excitatory strength is increased, the number of attractors increased as well (*Figure 9E*). In this regime, some areas are endowed with sufficient local reverberation to sustain persistent activity even when decoupled from the rest of the system; therefore, the importance of long-range coupling is diminished and a greater variety of attractor states is enabled. This can be understood by a simple example of two areas 1 and 2, each capable of two stimulus-selective persistent activity states; even without coupling, there are $2 \times 2 = 4$ attractor states with elevated firing. Thus, local and long-range connection strength have opposite effects on the number of attractors.

The cell type-specific input strength predicted firing rates across many attractors. In an example parameter regime ($\mu_{EE}$=0.04 nA and $g_{E,self} = 0.44$ nA), we identified 143 attractors. We correlated the input strength and cell type-specific input strength with the many attractor firing rates (*Figure 9F*). The raw input strength is weakly correlated with activity patterns. The cell type-specific input strength

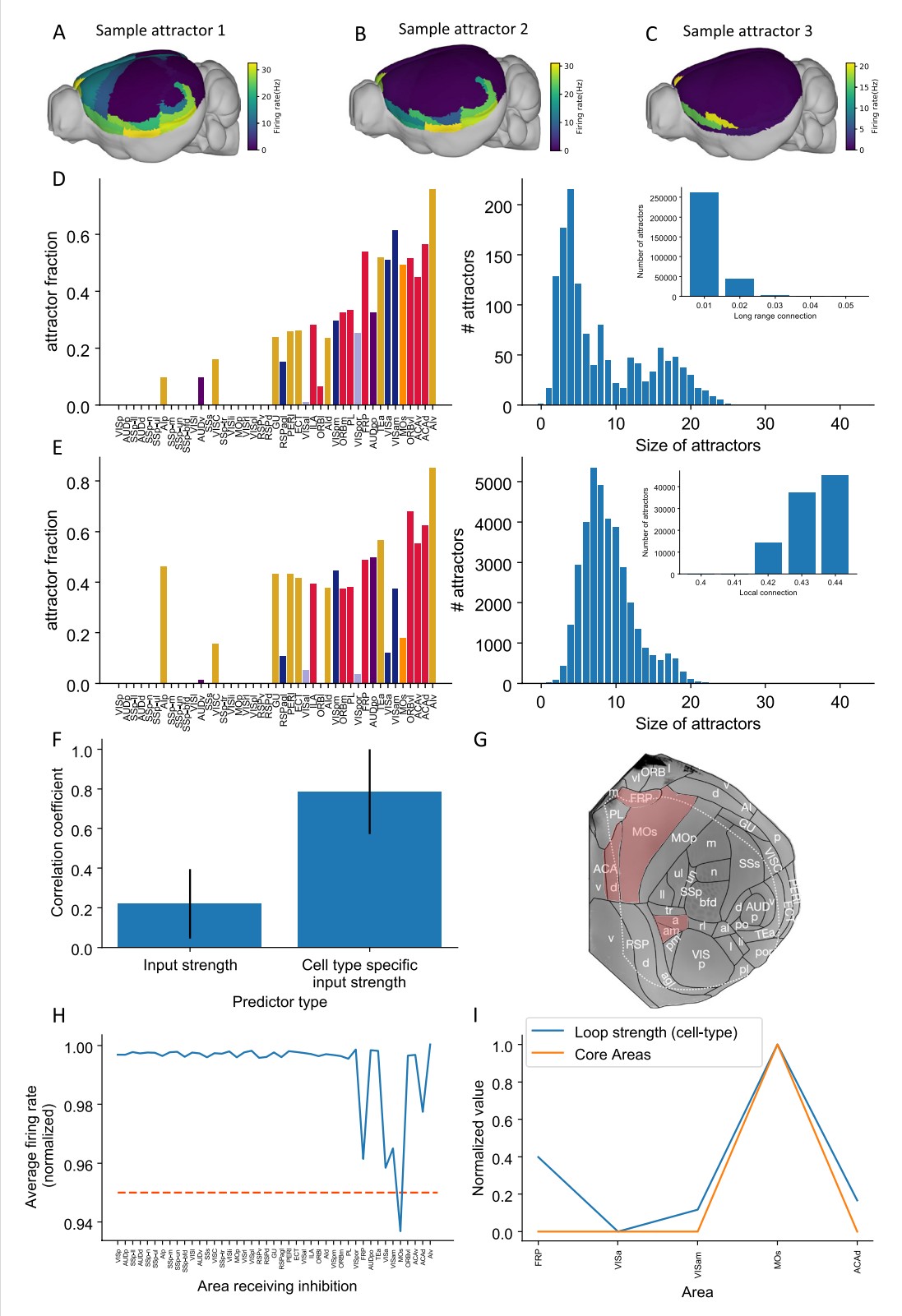

**Figure 9.** Multiple attractors coexist in the mouse working memory network. (**A–C**) Example attractor patterns with a fixed parameter set. Each attractor pattern can be reached via different external input patterns applied to the brain network. Delay activity is shown on a 3D brain surface. Color represents the firing rate of each area. (**D, E**) The distribution of attractor fractions (left) and number of attractors as a function of size (right) for different parameter combinations are shown. Attractor fraction of an area is the ratio between the number of attractors that include the area and the total number of

*Figure 9 continued*

identified attractors. In (**D**), local excitatory strengths are fixed ($g_{E,self}$=0.44 nA) while long-range connection strengths vary in the range $\mu_{EE}$ = 0.01–0.05 nA. Left and right panels of (**D**) show one specific parameter $\mu_{EE}$ = 0.03 nA. Inset panel of (**D**) shows the number of attractors under different long-range connection strengths while $g_{E,self}$ is fixed at 0.44 nA. In (**E**), long-range connection strengths are fixed ($\mu_{EE}$=0.02 nA) while local excitatory strengths varies in the range $g_{E,self}$ = 0.4–0.44 nA. Left and right panels of (**E**) show one specific parameter $g_{E,self}$ = 0.43 nA. Inset panel of (**E**) shows the number of attractors under different local excitatory strengths, while $\mu_{EE}$ is fixed at 0.02 nA. (**F**) Prediction of the delay-period firing rate using input strength and cell type-specific input strength for each attractor state identified under $\mu_{EE}$ = 0.04 nA and $g_{E,self}$ = 0.44 nA. A total of 143 distinct attractors were identified and the average correlation coefficient using cell type-specific input strength is better than that using input strength. (**G**) A example attractor state identified under the parameter regime $\mu_{EE}$ = 0.03 nA and $g_{E,self}$ = 0.44 nA. The five areas with persistent activity are shown in red. (**H**) Effect of single area inhibition analysis for the attractor state in (**G**). For a regime where five areas exhibit persistent activity during the delay period, inactivation of the premotor area MOs yields a strong inhibition effect (<0.95 orange dashed line) and is therefore a core area for the attractor state in (**G**). (**I**) Cell type-specific loop strength (blue) is plotted alongside core areas (orange) for the attractor state in (**G**). Only five areas with persistent activity are used to calculate the loop strength. Loop strength is normalized to be within the range of 0 and 1. High cell type-specific loop measures predict that an area is a Core area (prediction accuracy is 100% correct). The number of areas is limited, so prediction accuracy is very high.

is strongly correlated with activity across attractors. This shows that the cell type-specific connectivity measures are better at predicting the firing rates in many scenarios. These results further prove the importance of having cell type-specific connectivity for modeling brain dynamics.

Different attractor states rely on distinct subsets of core areas. In one example attractor, we found five areas that show persistent activity: VISa, VISam, FRP, MOs, and ACAd (*Figure 9G*) (parameter regime, $\mu_{EE}$ = 0.03 nA and $g_{E,self}$ = 0.44 nA). We repeated the previous inhibition analysis to identify core areas for this attractor state. Inhibiting one area, MOs, during the delay had the strongest effect on delay activity in the other parts of the attractor (*Figure 9H*). MOs also showed strong persistent activity during delay period. This is consistent with its role in short-term memory and planning (*Li et al., 2015*; *Inagaki et al., 2019*). According to our definition, MOs is a core area for this attractor. To calculate a loop strength that was specific to this attractor, we only examined connections between these five areas. The cell type-specific loop strength was the strongest in area MOs (*Figure 9I*). Thus, we can identify likely core areas for individual attractor states from cell type-specific structural measures. This also demonstrates that different attractor states can be supported by distinct core areas.

## Discussion

We developed a connectome-based dynamical model of the mouse brain. The model was capable of internally maintaining sensory information across many brain areas in distributed activity in the absence of any input. To our knowledge, this is the first biologically based model of the entire mouse cortex and the thalamocortical system that supports a cognitive function, in this case working memory. Together with our recent work (*Mejías and Wang, 2022*; *Froudist-Walsh et al., 2021*; *Froudist-Walsh et al., 2023*), it provides an important reference point to study the differences between rodents and monkeys.

Our main findings are threefold. First, the mnemonic activity pattern is shaped by the differing densities of PV interneurons across cortical areas. Areas with a high PV cell fraction encoded information only transiently, while those with low PV cell fraction sustained activity for longer periods. Thus, the gradient of PV cells (*Kim et al., 2017*) has a definitive role in separating rapid information processing in sensory areas from sustained mnemonic information representation in associative areas of the mouse cortex. This is consistent with the view that each local area operates in the 'inhibition-stabilizing regime' where recurrent excitation alone would lead to instability but the local network is stabilized by feedback inhibition, which may arise from long-range excitatory inputs to inhibitory neurons. This is consistent with the regime of the primary visual cortex (*Douglas et al., 1995*; *Murphy and Miller, 2009*). Second, we deliberately considered two different dynamical regimes: when local recurrent excitation is not sufficient to sustain persistent activity and when it is. In the former case, distributed working memory must emerge from long-range interactions between parcellated areas. Thereby the concept of synaptic reverberation (*de NÓ, 1933*; *Wang, 2001*; *Wang, 2021*) is extended to the large-scale global brain. Note that currently it is unclear whether persistent neural firing observed in a delay dependent task is generated locally or depends on long-distance reverberation among multiple brain regions. Our work made the distinction explicit and offers specific predictions to

be tested experimentally. Third, presently available connectomic data are not sufficient to account for neural dynamics and distributed cognition, and we propose cell type-specific connectomic measures that are shown to predict the observed distributed working memory representations. Our model underscores that, although connectome databases are an invaluable resource for basic neuroscience, they should be supplemented with cell type-specific information.

We found that recurrent loops within the cortex and the thalamocortical network aided in sustaining activity throughout the delay period (*Guo et al., 2017*; *Schmitt et al., 2017*). The presence of thalamocortical connections had a similar effect on the model as cortico-cortical projections, with the distinct contributions of the thalamus to large-scale dynamics still to be uncovered (*Shine et al., 2018*; *Jaramillo et al., 2019*). The specific pattern of cortico-cortical connections was also critical to working memory. However, standard graph theory measures based on the connectome were unable to predict the pattern of working memory activity. By focusing on cell type-specific interactions between areas, we were able to reveal a core of cortical areas. The core is connected by excitatory loops and is responsible for generating a widely distributed pattern of sustained activity. This clarifies the synergistic roles of the connectome and gradients of local circuit properties in producing a distributed cognitive function.

Previous large-scale models of the human and macaque cortex have replicated functional connectivity (*Deco et al., 2014*; *Demirtaş et al., 2019*; *Honey et al., 2007*; *Schmidt et al., 2018*; *Shine et al., 2018*; *Cabral et al., 2011*; *Wang et al., 2019*) and propagation of information along the cortical hierarchy (*Chaudhuri et al., 2015*; *Joglekar et al., 2018*; *Diesmann et al., 1999*). More recently, large-scale neural circuit models have been developed specifically to reproduce neural activity during cognitive tasks (*Mejías and Wang, 2022*; *Froudist-Walsh et al., 2021*; *Klatzmann et al., 2022*). These models consider the fact that in the macaque cortex the density of spines on pyramidal cells increases along the cortical hierarchy (*Elston and Rosa, 1998*; *Elston, 2007*; *Chaudhuri et al., 2015*). In a large-scale model of the macaque cortex (*Chaudhuri et al., 2015*), it was shown that this 'excitatory gradient' (*Wang, 2020*) is correlated with the distribution of intrinsic timescales in the cortex (*Murray et al., 2014*) and is consistent with spatially distributed working memory patterns (*Mejías and Wang, 2022*; *Froudist-Walsh et al., 2021*). Such excitatory gradients based on spine count are less pronounced and may be entirely absent in the rodent cortex (*Ballesteros-Yáñez et al., 2010*; *Gilman et al., 2017*). However, there are gradients of synaptic inhibition in the mouse cortex (*Kim et al., 2017*; *Wang, 2021*). Kim et al. showed that the ratio of SST+ neurons to PV+ neurons is low for early sensory areas and motor areas, while it is high in association areas such as the frontal cortex. We have used this gradient of inhibition in our model to show that spatially distributed persistent-activity patterns in the mouse cortex do not require gradients of recurrent excitation. In our model, the PV gradient and CIB may be particularly important to maintain the stability of an otherwise highly excitable cortical area. Along these lines, we predict that local recurrency in the mouse early sensory areas is higher than in the primate. Consistent with this claim, both the spine density and the number of excitatory and inhibitory synapses in layer 2/3 pyramidal neurons in area V1 are higher in mouse compared to macaque (Figure 5A in *Gilman et al., 2017* and Figure 1A in *Wildenberg et al., 2021*).

Other anatomical properties at the area and single-cell level may be informative of the differences in computational and/or cognitive abilities between rodents and macaques. In the language of network theory, the macaque cortex is a densely connected graph at an inter-area level, with the connectivity spanning five orders of magnitude (*Markov et al., 2014b*), which is more than what is expected for small-world networks (*Bassett and Bullmore, 2017*). Critically, the mouse 'connectome' (e.g., *Oh et al., 2014*; *Harris et al., 2019*; *Knox et al., 2019*) has even denser area-to-area connections. In the visual cortex, individual neurons target more cortical areas in the mouse (*Siu et al., 2021*) and they have more inhibitory and excitatory synapses (*Wildenberg et al., 2021*). Thus, connectivity in the mouse is denser at both the area and single-cell levels (at least for primary visual cortex). We propose that there is a greater functional specialization in the primate cortex which is afforded by the sparser and more targeted patterns of connectivity at the single-cell and area levels. Other differences to explore in future computational models include the ratio of NMDA to AMPA-mediated synaptic currents, which is approximately constant in the mouse cortex (*Myme et al., 2003*) but varies along the cortical hierarchy in primates (*Yang et al., 2018*; *Klatzmann et al., 2022*), as well as hierarchy, which is defined based on feedforward and feedback projections in the mouse (*Harris et al., 2019*) and primate (*Markov et al., 2014a*; *Markov et al., 2014b*).

We found that traditional graph theory metrics of connectivity were unable to predict the working memory activity in the mouse brain. This may be due to the almost fully connected pattern of interareal connectivity in the mouse cortex (*Gămănut et al., 2018*). This implies that, qualitatively, all areas have a similar set of cortical connections. In our model, we allowed the cell type target of interareal connections to change according to the relative position of the areas along the cortical hierarchy. Specifically, feedforward connections had a greater net excitatory effect than feedback connections, a hypothesis which we refer to as CIB. This preferential targeting of feedback projections serves to stabilize the otherwise excitable activity of sensory areas (*Mejías and Wang, 2022*) and is consistent with recent experiments that report long-range recruitment of GABAergic neurons in early sensory areas (*Campagnola et al., 2022*; *Shen et al., 2022*; *Naskar et al., 2021*). Our model predicts that if there is a weak correlation between PV cell density and delay firing rate across cortical areas, then the CIB mechanism is at play. Moreover, the model results suggest that CIB is particularly important in the regime where local connections are not sufficient to sustain spatially patterned persistent activity. We also showed that there are parameter regimes where CIB becomes less important, provided there is a gradient of synaptic inhibition as in the mouse cortex (*Kim et al., 2017*, but see *Nigro et al., 2022*). Notably, the model's resilience to parameter variations in inhibitory connection strengths is significantly enhanced when both the PV gradient and CIB are present. Given that working memory is a fundamental cognitive function observed across many individual brains with anatomical differences, the inclusion of multiple inhibitory mechanisms that allow for connectivity variations might confer evolutionary advantages. Although there is some evidence for similar inhibitory gradients in humans (*Burt et al., 2018*) and macaque (*Torres-Gomez et al., 2020*), the computational consequences of differences across species remain to be established.

To conclude, the manner in which long-range recurrent interactions affect neural dynamics depends not only on the existence of excitatory projections per se, but also on the target neurons' cell type. Thus, for some cortical areas afferent long-range excitatory connections promote working memory-related activity while for some others, for example, early sensory areas, it does not. Moreover, the existence of long-range interactions is consistent with potentially distinct dynamical regimes. For example, in one regime some areas exhibit independent persistent activity, that is, local recurrent interactions are sufficient to sustain a memory state for these areas, while others do not. In this regime, CIB is not required for the existence of distributed persistent activity patterns. In another regime, none of the areas can sustain a memory state without receiving long-range input. These two regimes are functionally distinct in terms of their robustness to perturbation as well as in the number of attractors that they can sustain. These regimes may be identified via perturbation analysis in future experimental and theoretical work.

By introducing cell type-specific graph theory metrics, we were able to predict the pattern and strength of delay period activity with high accuracy. Moreover, we demonstrated how cell type-specific graph-theory measures can accurately identify the core subnetwork, which can also be identified independently using a simulated large-scale optogenetic experiment. We found a core subnetwork of areas that, when inhibited, caused a substantial drop in activity in the remaining cortical areas. This core working memory subnetwork included frontal cortical areas with well-documented patterns of sustained activity during working memory tasks, such as prelimbic (PL), infralimbic (ILA), and medial orbitofrontal cortex (ORBm) (*Schmitt et al., 2017*; *Liu et al., 2014*; *Wu et al., 2020*). However, the core subnetwork for the visual working memory task we assessed was distributed across the cortex. It also included temporal and higher visual areas, suggesting that long-range recurrent connections between the frontal cortex and temporal and visual areas are responsible for generating persistent activity and maintaining visual information in working memory in the mouse.

Some of the areas that were identified as core areas in our model have been widely studied in other tasks. For example, the gustatory area exhibits delay-period preparatory activity in a taste-guided decision-making task and inhibition this area during the delay period impairs behavior (*Vincis et al., 2020*).

The core visual working memory subnetwork generates activity that is then inherited by many readout areas, which also exhibit persistent activity. However, inhibiting readout areas only mildly affects the activity of other areas (*Figure 7*, *Figure 7—figure supplement 1*). The readout areas in our model were a mixture of higher visual areas, associative areas, and premotor areas of cortex. We also concluded that MOs is a readout area and not a core area. This finding may be surprising considering previous studies that have shown this area to be crucial for short-term memory maintenance, planning, and movement

execution during a memory-guided response task (*Guo et al., 2017*; *Guo et al., 2014*; *Inagaki et al., 2019*; *Li et al., 2015*; *Wu et al., 2020*; *Voitov and Mrsic-Flogel, 2022*). This task has shown to engage, not only ALM, but a distributed subcortical-cortical network that includes the thalamus, basal ganglia, and cerebellum (*Svoboda and Li, 2018*). We note that in the version of the memory-guided response task studied by Svoboda and others, short-term memory is conflated with movement preparation. In our task, we proposed to study the maintenance of sensory information independent of any movement preparation as in delayed match-to-sample tasks and variations thereof. It is for this behavioral context that we found that MOs is not a core visual working memory area. We emphasize that readout areas are not less important than core areas as readout areas can use the stored information for further computations and thus some readout areas are expected to be strongly coupled to behavior. Indeed, there is evidence for a differential engagement of cortical networks depending on the task design (*Jonikaitis et al., 2023*) and on effectors (*Kubanek and Snyder, 2015*). If ALM is indeed a readout area for sensory working memory tasks (e.g., *Schmitt et al., 2017*), then the following prediction arises. Inhibiting ALM should have a relatively small effect on sustained activity in core areas (such as PL) during the delay period. In contrast, inhibiting PL and other core areas may disrupt sustained activity in ALM. Even if ALM is not part of the core for sensory working memory, it could form part of the core for motor preparation tasks (*Figure 9G*). We found a high cell type-specific loop strength for area ALM, like that in core areas, which supports this possibility (*Figure 9I*). Furthermore, we found some attractor states for which the MOs was classified as a core area that do not contain area PL. This result is supported by a recent study that found no behavioral effect after PL inhibition in a motor planning task (*Wang et al., 2021*). Therefore, the core subnetwork required for generating persistent activity is likely task-dependent. Future modeling work may help elucidate the biological mechanisms responsible for switching between attractor landscapes for different tasks.

Neuroscientists are now observing task-related neural activity at single-cell resolution across much of the brain (*Stringer et al., 2019*; *Steinmetz et al., 2019*). This makes it important to identify ways to distinguish the core areas for a function from those that display activity that serves other purposes. We show that a large-scale inhibition protocol can identify the core subnetwork for a particular task. We further show how this core can be predicted based on the interareal loops that target excitatory neurons. Were such a cell type-specific interareal connectivity dataset available, it may help interpretation of large-scale recording experiments. This could also focus circuit manipulation on regions most likely to cause an effect on the larger network activity and behavior. Our approach identifies the brain areas that work together to support working memory. It also identifies those that benefit from such activity to serve other purposes. Our simulation and theoretical approach is therefore ideally suited to understand the large-scale anatomy, recording, and manipulation experiments which are at the forefront of modern systems neuroscience.

Neuroscience has rapidly moved into a new era of investigating large-scale brain circuits. Technological advances have enabled the measurement of connections, cell types, and neural activity across the mouse brain. We developed a model of the mouse brain and theory of working memory that is suitable for the large-scale era. Previous reports have emphasized the importance of gradients of dendritic spine expression and interareal connections in sculpting task activity in the primate brain (*Mejías and Wang, 2022*; *Froudist-Walsh et al., 2021*). Although these anatomical properties from the primate cortex are missing in the mouse brain (*Gămănut et al., 2018*; *Gilman et al., 2017*), other properties such as interneuron density (*Kim et al., 2017*) may contribute to areal specialization. Indeed, our model clarifies how gradients of interneurons and cell type-specific interactions define large-scale activity patterns in the mouse brain during working memory, which enables sensory and associative areas to have complementary contributions. Future versions of the large-scale model may consider different interneuron types to understand their contributions to activity patterns in the cortex (*Kim et al., 2017*; *Tremblay et al., 2016*; *Froudist-Walsh et al., 2021*; *Meng et al., 2023*; *Wang et al., 2004*; *Nigro et al., 2022*), the role of interhemispheric projections in providing robustness for short-term memory encoding (*Li et al., 2016*), and the inclusions of populations with tuning to various stimulus features and/or task parameters that would allow for switching across tasks (*Yang et al., 2019*). Importantly, these large-scale models may be used to study other important cognitive computations beyond working memory, including learning and decision making (*Abbott et al., 2017*; *Abbott et al., 2020*).

# Materials and methods

## Anterograde tracing, connectivity data

We used the mouse connectivity map from the Allen Institute (*Oh et al., 2014*) to constrain our large-scale circuit model of the mouse brain. The Allen Institute measured the connectivity among cortical and subcortical areas using an anterograde tracing method. In short, they injected virus and expressed fluorescent protein in source areas and performed fluorescent imaging in target areas to measure the strength of projections from source areas. Unlike retrograde tracing methods used in other studies (*Markov et al., 2014b*), the connectivity strength measured using this method does not need to be normalized by the total input or output strength. This means that connectivity strength between any two areas is comparable.

The entries of the connectivity matrix from the Allen Institute can be interpreted as proportional to the total number of axonal fibers projecting from unit volume in one area to unit volume in another area. Before incorporating the connectivity into our model, we normalized the data as follows. In each area, we model the dynamics of an 'average' neuron, assuming that the neuron receive inputs from all connected areas. Thus, we multiplied the connectivity matrix by the volume $Vol_j$ of source area $j$ and divided by the average neuron density $d_i$ in target area $i$:

$$W_{norm,ij} = W_{raw,ij} \frac{Vol_j}{d_i} \tag{1}$$

where $W_{raw,ij}$ is the raw, that is, original, connection strength from unit volume in source area $j$ to unit volume in target area $i$, $Vol_j$ is the volume of source area $j$ (*Wang et al., 2020*), and $d_i$ is the neuron density in source area $i$ (*Erö et al., 2018*). $W_{norm,ij}$ is the matrix that we use to set the long-range connectivity in our circuit model. We can define the corticothalamic connectivity $W_{ct,norm,ij}$ and thalamocortical connectivity $W_{tc,norm,ij}$ in a similar manner, except that we did not apply the normalization to thalamic connectivity due to not having enough neuron density data.

## Interneuron density along the cortex

Kim and colleagues measured the density of typical interneuron types in the brain (*Kim et al., 2017*). They expressed fluorescent proteins in genetically labeled interneurons and counted the number of interneurons using fluorescent imaging. We took advantage of these interneuron density data and specifically used the PV cell fraction to set local and long-range inhibitory weights.

The PV cell density of all layers is first divided by the total neuron density $d_i$ in the area $i$ to give the PV cell fraction $PV_{raw,i}$, which better reflects the expected amount of synaptic inhibition mediated by PV neurons. The PV cell fraction is then normalized across the whole cortex.

$$PV_i = \frac{PV_{raw,i} - min(PV_{raw,i})}{max(PV_{raw,i}) - min(PV_{raw,i})} \tag{2}$$

$PV_{raw,i}$ is the PV cell fraction in area $i$, and $PV_i$ is the normalized value of $PV_{raw}$, which will be used in subsequent modeling.

## Hierarchy in the cortex

The concept of hierarchy is important for understanding the cortex. Hierarchy can be defined based on mapping corticocortical long-range connections onto feedforward or feedback connections (*Felleman and Van Essen, 1991*; *Markov et al., 2014b*; *Harris et al., 2019*). Harris and colleagues measured the corticocortical projections and target areas in a series of systematic experiments in mice (*Harris et al., 2019*). Projection patterns were clustered into multiple groups and the label 'feedforward' or 'feedback' was assigned to each group. Feedforward and feedback projections were then used to determine relative hierarchy between areas. For example, if the projections from area A to area B are mostly feedforward, then area B has a higher hierarchy than area A. This optimization process leads to a quantification of the relative hierarchy of cortical areas $h_{raw,i}$. We defined the normalized hierarchy value $h_i$ as

$$h_i = \frac{h_{raw,i} - min(h_{raw,i})}{max(h_{raw,i}) - min(h_{raw,i})} \tag{3}$$

where $h_{raw,i}$ is the raw, that is, original hierarchical ordering from *Harris et al., 2019*. Due to data acquisition issues, six areas did not have a hierarchy value assigned to them (SSp-un, AUDv, GU, VISC, ECT, PERI) (*Harris et al., 2019*). We estimated hierarchy through a weighted sum of the hierarchy value of 37 known areas, while the weight is determined through the connectivity strength. The parameters $\alpha_h$ and $\beta_h$ are selected so that $h_{i,estimate}$ are close to $h_i$ for areas with known hierarchy.

$$h_{i,estimate} = \alpha_h \frac{\sum_{j=1}^{37} W_{raw,ij} h_j}{\sum_{j=1}^{37} W_{raw,ij}} + \beta_h \tag{4}$$

For the thalamocortical model, we also used the hierarchy value for thalamic areas (*Harris et al., 2019*). The hierarchy of thalamic areas is comparable to cortical areas, so in order to use it in the model, we also normalized them.

$$h_{th,i} = \frac{h_{th,raw,i} - min(h_{raw,i})}{max(h_{raw,i}) - min(h_{raw,i})} \tag{5}$$

To estimate the hierarchy value of thalamic areas with missing values, we used the known hierarchy value of the thalamic area next to the missing one as a replacement.

## Description of the local circuit

Our large-scale circuit model includes 43 cortical areas. Each area includes two excitatory populations, labeled *A* and *B*, and one inhibitory population, *C*. The two excitatory populations are selective to different stimuli. The synaptic dynamics between populations are based on previous firing rate models of working memory (*Wang, 1999*; *Wong and Wang, 2006*). The equations that define the dynamics of the synaptic variables are

$$\frac{dS_A}{dt} = -\frac{S_A}{\tau_N} + \gamma(1 - S_A)r_A \tag{6}$$

$$\frac{dS_B}{dt} = -\frac{S_B}{\tau_N} + \gamma(1 - S_B)r_B \tag{7}$$

$$\frac{dS_C}{dt} = -\frac{S_C}{\tau_G} + \gamma_I r_C \tag{8}$$

where $S_A$ and $S_B$ are the NMDA synaptic variables of excitatory populations *A* and *B*, while $S_C$ is the GABA synaptic variable of the inhibitory population *C*. $r_A$, $r_B$, and $r_C$ are the firing rates of populations *A*, *B*, and *C*, respectively. $\tau_N$ and $\tau_G$ are the time constants of NMDA and GABA synaptic conductances. $\gamma$ and $\gamma_I$ are the parameters used to scale the contribution of presynaptic firing rates. The total currents received $I_i$ ($i = A, B, C$) are given by

$$I_A = g_{E,self}S_A + g_{E,cross}S_B - g_{EI}S_C + I_{0A} + I_{LR,A} + x_A(t) \tag{9}$$

$$I_B = g_{E,self}S_B + g_{E,cross}S_A - g_{EI}S_C + I_{0B} + I_{LR,B} + x_B(t) \tag{10}$$

$$I_C = g_{IE}S_A + g_{IE}S_B - g_{II}S_C + I_{0C} + I_{LR,C} + x_C(t) \tag{11}$$

In these equations, $g_{E,self}$, $g_{E,cross}$ denote the connection strength between excitatory neurons with same or different selectivity, respectively. These parameters can be found in *Table 2*. These connection strengths are the same for different areas since there is no significant gradient for excitatory strength in mice. $g_{IE}$ are the connection strengths from excitatory to inhibitory neurons, while $g_{EI}$ and $g_{II}$ are connection strengths from inhibitory to excitatory neurons and from inhibitory to inhibitory neurons, respectively. These connections will be scaled by PV cell fraction $PV_i$ in the corresponding area. We will discuss the details in the next section. $I_{0i}$ ($i = A, B, C$) are constant background currents to each population. $I_{LR,i}$ ($i = A, B, C$) are the long-range (LR) currents received by each population. The term $x_i(t)$ where $i = A, B, C$ represents noisy contributions from neurons external to the network. It is modeled as an Ornstein–Uhlenbeck process:

**Table 2.** Parameters for numerical simulations.

| Parameter | Description | Task/figure | Value |
|---|---|---|---|
| | Cortical circuit parameters | | |
| $\tau_{NMDA}$ | NMDA synapse time constant | All figures | 60 ms |
| $\tau_{GABA}$ | GABA synapse time constant | All figures | 5 ms |
| $\tau_{AMPA}$ | AMPA synapse time constant | All figures | 2 ms |
| $\tau_{rates}$ | Neuron time constant | All figures | 2 ms |
| $\tau_{noise}$ | Noise time constant | All figures | 2 ms |
| $a, b, d$ | Parameters in excitatory F–I curve | All figures | 140 Hz/nA, 54 Hz, 308ms |
| $g_I, c_1, c_0, r_{0I}$ | Parameters in inhibitory F–I curve | All figures | 4, 615 Hz/nA, 177 Hz, 5.5 Hz |
| $\gamma$ | Parameters in NMDA excitatory synaptic equations | All figures | 1.282 |
| $\gamma_I$ | Parameters in GABA synaptic equations | All figures | 2 |
| $\gamma_A$ | Parameters in AMPA excitatory synaptic equations | All figures | 2 |
| $g_{E,self}$ | Local self-excitatory connections | **Figures 1–8** | 0.4 nA |
| $g_{E,cross}$ | Local cross-population excitatory connections | All figures | 10.7 pA |
| $g_{IE}$ | Local E to I connections | All figures | 0.2656 nA |
| $g_{EI,0}, g_{EI,scaling}$ | Local I to E connection strength | All figures | 0.192 nA, 0.83 |
| $g_{II,0}, g_{II,scaling}$ | Local I to I connection strength | All figures | 0.105 nA, 0.714 |
| $I_{0A}, I_{0B}$ | Background current for excitatory neurons | All figures | 0.305 nA |
| $I_{0C}$ | Background current for inhibitory neurons | All figures | 0.26 nA |
| $\sigma_A, \sigma_B$ | Standard deviation of excitatory noise current | All figures | 5 pA |
| $\sigma_C$ | Standard deviation of inhibitory noise current | All figures | 0 pA |
| $r_{0E}$ | Initial firing rate for excitatory neurons | All figures | 5 Hz |
| $r_{0I}$ | Initial firing rate for inhibitory neurons | All figures | 5.5 Hz |
| $\mu_{EE}$ | Long-range E to E connection strength | **Figures 1–4, 6–8** | 0.1 nA |
| $\mu_{IE}$ | Long-range E to I connection strength | **Figures 1–4, 6–8** | 0.167 nA |
| $\beta$ | Parameters in $m_{ij}$ | All figures | 2.42 |
| $k_{scale}$ | Parameters for scaling the connectivity matrix | All figures | 0.3 |
| $\alpha_h, \beta_h$ | Parameters for estimation of hierarchy | All figures | 1.33,  -0.22 |
| $I_{stim}$ | External stimulus strength | All figures | 0.5 nA |
| $I_{inh}$ | External input to inhibitory neurons | All figures | 5 nA |
| $T_{on}$ | Stimulus start time | All figures | 2 s |
| $T_{off}$ | Stimulus end time | All figures | 2.5 s |
| $T_{trial}$ | Simulation time for each trial | All figures | 10 s |
| $dt$ | Simulation time step | All figures | 0.5ms |
| | Thalamocortical network | | |
| $\mu_{EE}$ | Long-range E to E connection strength | **Figure 5** | 0.01 nA |

*Table 2 continued on next page*

*Table 2 continued*

| Parameter | Description | Task/figure | Value |
|---|---|---|---|
| $\mu_{IE}$ | Long-range $E$ to $I$ connection strength | *Figure 5* | 0.0167 nA |
| $g_{ct}$ | Corticothalamic connections strength | *Figure 5* | 0.32 nA |
| $g_{E,tc}$ | Thalamocortical connections to excitatory neurons | *Figure 5* | 0.6 nA |
| $g_{I,tc}$ | Thalamocortical connections to inhibitory neurons | *Figure 5* | 1.38 nA |
| | Simulation of multiple attractors | | |
| $\mu_{EE}$ | Long-range $E$ to $E$ connection strength | *Figure 9* | 0.01, 0.02, 0.03, 0.04, 0.05 nA |
| $\mu_{IE}$ | Long-range $E$ to $I$ connection strength | *Figure 9* | 0.0167, 0.033, 0.05, 0.066, 0.083 nA |
| $g_{E,self}$ | Local self-excitatory connections | *Figure 9* | 0.4, 0.41, 0.2, 0.43, 0.44 nA |

$$\tau_{noise}\frac{dx_i}{dt} = -x_i + \sqrt{\tau_{noise}}\sigma_i\zeta_i(t) \tag{12}$$

where $\zeta_i(t)$ is Gaussian white noise, $\tau_{noise}$ describes the time constant of external AMPA synapses, and $\sigma_i$ sets the strength of the noise for each population. $\sigma_A = \sigma_B = 5pA$ while $\sigma_C = 0pA$.

The steady state firing rate of each population is calculated based on a transfer function $\phi_i(I)$ of input current received by each population $I_i$ ($i = A, B, C$) given by

$$\phi_{A,B}(I_{A,B}) = \frac{aI_{A,B} - b}{1 - exp[-d(aI_{A,B} - b)]} \tag{13}$$

$$\phi_C(I_C) = [\frac{1}{g_I}(c_1 I - c_0) + r_0]^+ \tag{14}$$

Note that the transfer functions $\phi_i(t)$ are the same for two excitatory populations. $x^+$ denotes the positive part of the function $x$. The firing rate of each population follows equations

$$\tau_r\frac{dr_{A,B}}{dt} = -r_{A,B} + \phi_{A,B}(I_{A,B}) \tag{15}$$

$$\tau_r\frac{dr_C}{dt} = -r_C + \phi_C(I_C) \tag{16}$$

## Interneuron gradient and local connections

We scaled local interneuron connectivity with the interneuron density that was obtained using fluorescent labeling (*Kim et al., 2017*). Specifically, local I–I connections and local I–E connections are scaled by the interneuron density by setting the connection strength $g_{k,i}(k = EI, II)$ as a linear function of PV cell fraction $PV_i$ in area $i$.

$$g_{EI,i} = g_{EI,0}(1 + g_{EI,scaling}PV_i) \tag{17}$$

$$g_{II,i} = g_{II,0}(1 + g_{II,scaling}PV_i) \tag{18}$$

where $g_{k,0}$ ($k = EI, II$) is the base value of I to E connections and $g_{k,scaling}$ ($k = EI, II$) is the scaling factor of PV value. $g_{k,0}$ also accounts for the inhibition of other cell types not explicitly considered in this study.

## Hierarchy and long-range connections

Long-range (LR) connections between areas are scaled by connectivity data from the Allen Institute (*Oh et al., 2014*). We consider long-range connections that arise from excitatory neurons because most long-range connections in the cortex correspond to excitatory connections (*Petreanu et al., 2009*). Long-range connections will target excitatory populations in other brain areas with the same selectivity (*Zandvakili and Kohn, 2015*) and will also target inhibitory neurons. These long-range connections are given by the following equations:

$$I_{A,B,LR,i} = \Sigma_j \mu_{EE} W_{E,ij} S_{A,B,j} \tag{19}$$

$$I_{C,LR,i} = \Sigma_j \mu_{IE} W_{I,ij} (S_{A,j} + S_{B,j}) \tag{20}$$

where $W_E$ is the normalized long-range connectivity to excitatory neurons, and $W_I$ is the normalized long-range connectivity to inhibitory neurons. $\mu_{EE}$ and $\mu_{IE}$ are coefficients scaling the long-range E to E and E to I connection strengths, respectively.

Here, we assume that the long-range connections will be scaled by a coefficient that is based on the hierarchy of source and target area. To quantify the difference between long-range feedforward and feedback projections, we introduce $m_{ij}$ to measure the 'feedforwardness' of projections between two areas. According to our assumption of CIB, long-range connections to inhibitory neurons are stronger for feedback connections and weaker for feedforward connections, while the opposite holds for long-range connections to excitatory neurons. Following this hypothesis, we define $m_{ij}$ as a sigmoid function of the difference between the hierarchy value of source and target areas. For feedforward projections, $m_{ij} > 0.5$; for feedback projections, $m_{ij} < 0.5$. Excitatory and inhibitory long-range connection strengths are implemented by multiplying the long-range connectivity strength $W_{ij}$ by $m_{ij}$ and $(1 - m_{ij})$, respectively:

$$m_{ij} = \frac{1}{1 + e^{-\beta(h_i - h_j)}} \tag{21}$$

$$W_{E,ij} = m_{ij} W_{ij} \tag{22}$$

$$W_{I,ij} = (1 - m_{ij}) W_{ij} \tag{23}$$

with

$$W_{scale,ij} = (W_{norm,ij})^{k_{scale}} \tag{24}$$

$$W_{ij} = \frac{W_{scale,ij}}{max(W_{scale,ij})} \tag{25}$$

The connectivity $W_{norm,ij}$ is then rescaled to translate the broad range of connectivity values (over five orders of magnitude) to a range more suitable for our firing rate models. $k_{scale}$ is the coefficient used for this scaling. $k_{scale} < 1$ effectively makes the range much smaller than the original normalized connectivity $W_{norm,ij}$. After that, the scaled connectivity $W_{scale,ij}$ is then normalized so that the maximum value is fixed at 1.

## Simulations of replacing the PV gradient and CIB

In order to demonstrate the importance of PV gradient and CIB, we replace the PV gradient value/CIB with the average value accordingly in the simulation. Specifically, we replace PV gradient with the average PV cell fraction.

$$PV_{mean} = \frac{\sum_i PV_i}{n_{areas}} \tag{26}$$

$$g_{EI,i} = g_{EI,0}(1 + g_{EI,scaling} PV_{mean}) \tag{27}$$

$$g_{II,i} = g_{II,0}(1 + g_{II,scaling} PV_{mean}) \tag{28}$$

We also replace CIB with its average value 0.5, which means there is no bias to inhibitory cells for all long-range connections.

$$m_{ij} = 0.5 \tag{29}$$

$$W_{E,ij} = 0.5 W_{ij} \tag{30}$$

$$W_{I,ij} = 0.5 W_{ij} \tag{31}$$

For the simulations of varying the local inhibitory connection strengths, we specifically change the value of $g_{EI,0}$ and $g_{EI,scaling}$ for $g_{EI,i}$. For each combination of parameters of $g_{EI,0}$ and $g_{EI,scaling}$, simulations are performed for default parameters (no changes to PV gradient or CIB), PV average (PV gradient is replaced by average value), and CIB average (CIB is replaced by average value). The

average firing rate of all areas and number of areas showing persistent activity are quantified for each parameter combination.

In other simulations, we varied the parameters $g_{EI,0}$ and $g_{EI,scaling}$ with long-range connections $\mu_{EE}$ and $\mu_{IE}$ set to be 0. This enabled us to discover the range of parameter values for which individual areas were capable of maintaining persistent activity without input from other areas. In practice, the only key parameter that determines this behaviour is the smallest inhibitory connection strength of any area, $g_{EI,i} = g_{EI,0}$.

## Simulations and theoretical calculation of the baseline stability of the network

In the simulation focusing on the stability of the baseline state of the network, there was no external input provided to any of the areas apart from noise (*Equation 12*). The steady firing rate of each area after 10 s is recorded as a measure of the baseline stability.

We tested the baseline stability on five different scenarios (*Figure 4A and B*): In (1) and (2), we set the long-range connections $\mu_{EE}$ and $\mu_{IE}$ to zero since we focus on the local network. In (2), we also set the local inhibitory connections $g_{EI,0}$ to zero. In (3)–(5), the long-range connections are intact. In (4), we set the long-range connection to inhibitory neurons $\mu_{IE}$ to zero. In (5), we set the local inhibitory connections $g_{EI,0}$ to zero.

We analytically calculated the stability of baseline state for a local circuit when the long-range connections $\mu_{EE}$ and $\mu_{IE}$ are set to zero, which means $I_{LR,A}$, $I_{LR,B}$, $I_{LR,C}$ are zero in *Equations 9–11*. In *Equations 15 and 16*, we assume that $r_A$, $r_B$, and $r_C$ reach their steady states instantaneously since its time constant $\tau_r$ is much smaller than the time constant of NMDA synaptic variable $\tau_N$ in *Equations 6 and 7*. Thus, we can express the firing rate $r_A$, $r_B$, and $r_C$ as functions of synaptic variables $S_A$, $S_B$, and $s_C$ (*Equations 13–16*):

$$r_A = \phi_A(g_{E,self}S_A + g_{E,cross}S_B - g_{EI}S_C + I_{0A}) \tag{32}$$

$$r_B = \phi_A(g_{E,self}S_B + g_{E,cross}S_A - g_{EI}S_C + I_{0A}) \tag{33}$$

$$r_C = \phi_C(g_{IE}S_A + g_{IE}S_B - g_{II}S_C + I_{0C}) \tag{34}$$

where $\phi_A$ and $\phi_C$ have the same form as *Equations 13 and 14*.

Then we can insert *Equations 32–34* into *Equations 6–8* to obtain a differential equation for $S_A$, $S_B$, and $S_C$.

$$\frac{dS_A}{dt} = -\frac{S_A}{\tau_N} + \gamma(1 - S_A)\phi_A(g_{E,self}S_A + g_{E,cross}S_B - g_{EI}S_C + I_{0A}) \tag{35}$$

$$\frac{dS_B}{dt} = -\frac{S_B}{\tau_N} + \gamma(1 - S_B)\phi_A(g_{E,self}S_B + g_{E,cross}S_A - g_{EI}S_C + I_{0A}) \tag{36}$$

$$\frac{dS_C}{dt} = -\frac{S_C}{\tau_G} + \gamma_I\phi_C(g_{IE}S_A + g_{IE}S_B - g_{II}S_C + I_{0C}) \tag{37}$$

The steady state of $S_A$, $S_B$, and $S_C$ can be solved numerically by setting the left side of the above equations to be zero. We denote the right side of the equations as *FA*, *FB*, and *FC*. Then we can calculate the Jacobian matrix and its eigenvalues.

$$J_{S_A,S_B,S_C} = \begin{bmatrix} \dfrac{dFA}{dS_A} & \dfrac{dFA}{dS_B} & \dfrac{dFA}{dS_C} \\ \dfrac{dFB}{dS_A} & \dfrac{dFB}{dS_B} & \dfrac{dFB}{dS_C} \\ \dfrac{dFC}{dS_A} & \dfrac{dFC}{dS_B} & \dfrac{dFC}{dS_C} \end{bmatrix} \tag{38}$$

If the real part of all the eigenvalues is negative, then that means the baseline state is stable. The eigenvalues of the scenario (1) are −10.4, −12.5, and −229.8, while those of scenario (2), where local inhibitory connections $g_{EI,0}$ are zero, are −7.4, −7.9, −232.3. These results coincide with the simulation results of *Figure 4A*.

We also considered an alternative parameter regime, where the local excitatory connections $g_{E,self}$ is set to a higher level $g_{E,self} = 0.6\,\mathrm{nA}$. The local inhibitory connections strength $g_{EI,0}$ is also set to a

higher level $g_{EI,0} = 0.5$ nA to balance the increased excitatory connections. Under such alternative parameter regime, we performed similar analysis as the five different scenarios in *Figure 4A and B*. The results are shown in *Figure 4C and D*. In simulations of a network with intact long-range connections and increased local excitatory connections (in *Figure 4D* and also in *Figure 4F*), we changed the long-range connections strength $\mu_{EE} = 0.19$ nA. In *Figure 4C*, when we gradually decrease the inhibitory connection strength $g_{EI,0}$ from 0.5 nA to 0 (from blue dots to orange dots), analytical calculations demonstrate that the stable low firing rate state disappears via a saddle node bifurcation at $g_{EI,0} = 0.175$ nA (for area AIp). This demonstrates that, upon removal of inhibition, the high firing rate in *Figure 4C* corresponds to a distinct state and not simply a shift of the baseline state.

In the increased local excitatory connection regime, we further introduced temporary external input to each local brain areas and record its stable firing rate shown in *Figure 4E*.

In the simulation of *Figure 4F*, we used the classic simulation protocol: an temporary external input is given to primary visual cortex and the delay-period firing rate of each areas are recorded and shown.

## Thalamocortical network model

### Corticothalamic connectivity

We introduced thalamic areas in the network to examine their effect on cortical dynamics. Each thalamic area includes two excitatory populations, *A* and *B*, with no inhibitory population. These two populations share the same selectivity with the corresponding cortical areas. Unlike cortical areas, there are no recurrent connections between thalamic neurons (*Sherman, 2007*). Thalamic currents have the following contributions (tc stands for thalamocortical connections and ct for corticothalamic connections):

$$I_{th,A,B} = I_{ct,A,B} + I_{th,0,A,B} + I_{th,noise,A,B} \tag{39}$$

where $I_{th,i}$ ($i = A, B$) is the total current received by each thalamic population, $I_{ct,i}$ ($i = A, B$) is the long-range current from cortical areas to target thalamic area, $I_{th,0,i}$ ($i = A, B$) is the background current for each population, and $I_{th,noise,i}$ ($i = A, B$) is the noise input to thalamic population *A* and *B*, which we set to 0 in our simulations. $I_{ct,i}$ ($i = A, B$) has the following form:

$$I_{ct,A,B,i} = g_{ct} W_{ct,E,ij} S_{k,j} \tag{40}$$

where $W_{ct,E,ij}$ is the LR connectivity to thalamic neurons, and $S_{k,j}$ is the synaptic variable of population *k* ($k = A, B$) in cortical area *j*. Since all thalamic neurons are excitatory, we model corticothalamic projections as in the previous section:

$$m_{ct,ij} = \frac{1}{1 + e^{-\beta(h_{th,i} - h_j)}} \tag{41}$$

$$W_{ct,E,ij} = m_{ct,ij} W_{ct,ij} \tag{42}$$

$$W_{ct,I,ij} = (1 - m_{ct,ij}) W_{ct,ij} \tag{43}$$

where

$$W_{ct,scale,ij} = (W_{ct,norm,ij})^{k_{scale}} \tag{44}$$

$$W_{ct,ij} = \frac{W_{ct,scale,ij}}{max(W_{ct,scale,ij})} \tag{45}$$

$W_{ct,norm,ij}$ is the normalized connection strength from cortical area *j* to thalamic area *i*. $m_{ct,ij}$ is the coefficient quantifying how the long-range connections target excitatory neurons based on cortical hierarchy $h_j$ and thalamic hierarchy $h_{th,i}$.

The thalamic firing rates are described by

$$\tau_r \frac{dr_{th,A,B}}{dt} = -r_{th,A,B} + \phi_{th,A,B}(I_{th,A,B}) \tag{46}$$

with the activation function for thalamic neurons given by

$$\phi_{th,A,B}(I_{th,A,B}) = \frac{aI_{th,A,B} - b}{1 - exp[-d(aI_{th,A,B} - b)]} \tag{47}$$

Thalamic neurons are described by AMPA synaptic variables (*Jaramillo et al., 2019*):

$$\frac{dS_{th,A,B}}{dt} = -\frac{S_{th,A,B}}{\tau_A} + \gamma_A r_{th,A,B} \tag{48}$$

## Thalamocortical connectivity

The connections from thalamic neurons to cortical neurons follow these equations:

$$I_{tc,A,B,i} = g_{E,tc} W_{E,tc,ij} S_{th,A,B,j} \tag{49}$$

$$I_{tc,C,i} = g_{I,tc} W_{I,tc,ij} (S_{th,A,j} + S_{th,B,j}) \tag{50}$$

and connectivity

$$m_{tc,ij} = \frac{1}{1 + e^{-\beta(h_i - h_{th,j})}} \tag{51}$$

$$W_{E,tc,ij} = m_{tc,ij} W_{tc,ij} \tag{52}$$

$$W_{I,tc,ij} = (1 - m_{tc,ij}) W_{tc,ij} \tag{53}$$

and connectivity matrix

$$W_{tc,scale,ij} = (W_{tc,norm,ij})^{k_{scale}} \tag{54}$$

$$W_{tc,ij} = \frac{W_{tc,scale,ij}}{max(W_{tc,scale,ij})} \tag{55}$$

The thalamocortical input is added to the total input current of each cortical population.

$$I_A = g_{E,self} S_A + g_{E,cross} S_B + g_{EI} S_C + I_{0A} + I_{LR,A} + I_{tc,A} + x_A(t) \tag{56}$$

$$I_B = g_{E,self} S_B + g_{E,cross} S_A + g_{EI} S_C + I_{0B} + I_{LR,B} + I_{tc,B} + x_B(t) \tag{57}$$

$$I_C = g_{IE} S_A + g_{IE} S_B + g_{II} S_C + I_{0C} + I_{LR,C} + I_{tc,C} + x_C(t) \tag{58}$$

## Calculation of network structural measures

We considered three types of structural measures. The first one is input strength. Input strength of area $i$ is the summation of the connection strengths onto node $i$. It quantifies the total external input onto area $i$.

$$W_{input,i} = \sum_{j=1}^{n} W_{ij} \tag{59}$$

The second one is eigenvector centrality (*Newman, 2018*). Eigenvector centrality of area $i$ is the $i$th element of the leading eigenvector of the connectivity matrix. It quantifies how many areas are connected with the target area $i$ and how important these neighbors are. $W$ is a matrix where each element is $W_{ij}$.

$$W = Q \Lambda Q^{-1} \tag{60}$$

$$C_{eig,i} = q_{i1} \tag{61}$$

The third structural measure is loop strength, which quantifies how each area is involved in strong recurrent loops. We first define the strength of a single loop $k$

$$L_k = \prod_{A_i, A_j \in loop_k} W_{ij} \tag{62}$$

and then the loop strength $S_{A_i}$ of a single area $A_i$

$$S_{A_i} = \sum_{A_i \in loop_k} L_k \tag{63}$$

We now focus on cell type-specific structural measures. Cell type specificity is introduced via a coefficient $k_{cell}$ that scales all long-range connection strengths (cell type projection coefficient):

$$k_{cell} = m_{ij} - PV_i(1 - m_{ij}) \tag{64}$$

Thus, we can define cell type-specific connectivity as

$$W_{cell,ij} = (m_{ij} - PV_i(1 - m_{ij}))W_{ij} \tag{65}$$

The cell type-specific connectivity is further normalized so that the maximum value is 1.

$$\tilde{W}_{ij} = \frac{W_{cell,ij}}{max(W_{cell,ij})} \tag{66}$$

and cell type-specific input strength could be defined as

$$W_{input,i,cellspec} = \sum_{j=1}^{n} \tilde{W}_{ij} \tag{67}$$

Similarly, cell type-specific eigenvector centrality is defined as

$$\tilde{W} = \tilde{Q}\tilde{\Lambda}\tilde{Q}^{-1} \tag{68}$$

$$C_{eig,i,cellspec} = \tilde{q}_{i1} \tag{69}$$

where $\tilde{W}$ is a matrix where each element is $\tilde{W}_{ij}$ and the cell type-specific loop strength is defined as

$$L_{k,cellspec} = \prod_{A_i,A_j \in loop_k} \tilde{W}_{ij} \tag{70}$$

$$S_{A_i,cellspec} = \sum_{A_i \in loop_k} L_{k,cellspec} \tag{71}$$

As a comparison, we also calculated the sign-only loop strength and no PV loop strength.

We can define sign-only connectivity as

$$W_{signonly,ij} = sgn(m_{ij} - (1 - m_{ij}))W_{ij} \tag{72}$$

where $sgn(x)$ is the sign function, which returns positive or negative values based on the sign of $x$. The major difference between sign-only connectivity and cell type-specific connectivity is that the strength of long-range projection bias are not considered except the sign of it.

We can also define noPV connectivity as:

$$W_{noPV,ij} = (m_{ij} - (1 - m_{ij}))W_{ij} \tag{73}$$

The difference between noPV connectivity and cell type-specific connectivity is that the different strengths of local connections for each area are not considered in the noPV connectivity.

We also used the sign-only and noPV variants of connectivity measures to predict the delay-period firing rate and classify core areas. This enabled us to compare these simplified measures to the cell type-specific connectivity measures.

## Stimulation protocol and inhibition analysis

The model is simulated using an stochastic differential equation solver: Euler–Maruyama method. We write customized program using Python to implement this numerical method. The time step is set to be $dt$, and all the firing rates, synaptic variables, and currents are initialized to be zero.

We simulate a working memory task by applying an external current $I_{stim}$ to one of the excitatory populations, which represents a sensory (e.g., visual) stimulus that is to be remembered across a delay period. The external current is a pulsed input with start time $T_{on}$ and offset time $T_{off}$. Without losing generality, we assume that the external input is provided to population $A$. In most of the simulations in this study, we simulate a visual working memory task, with the external input applied to VISp. The simulation duration is $T_{trial}$, and we used a time step of $dt$. The delay period is defined as the duration

between the offset time $T_{off}$ and trial end $T_{trial}$. In order to obtain a stable firing rate, the delay-period firing rate is calculated by averaging the firing rate from 2 s until the end of the delay period to 0.5 s until end. Firing rate, PV cell fraction, and hierarchy are plotted on a 3D brain surface using the website Scalable Brain Atlas (https://scalablebrainatlas.incf.org/index.php).

We apply inhibition analysis to understand the robustness of attractors and, more importantly, investigate which areas play an important role in maintaining the attractor state. Excitatory input was applied to the inhibitory population $I$ to simulate opto-genetic inhibition. The external input $I_{inh}$ is strong compared to $I_{stim}$ and results in an elevated firing rate of the inhibitory population, which in turn decreases the firing rate of the excitatory populations. Usually the inhibition is applied to a single area. When inhibition is applied during the stimulus period, its start and end times are equal to $T_{on}$ and $T_{off}$, respectively. When inhibition is applied during delay period, its start time is later than $T_{off}$ to allow the system settle to a stable state. Thus, the onset of inhibition starts 2 s after $T_{off}$ and lasts until the end of trial. In the case of thalamocortical network simulations, we inhibit thalamic areas by introducing a hyperpolarizng current to both excitatory populations since we do not have inhibitory populations in thalamic areas in the model.

To quantify the effect of single area or multiple areas inhibition, we calculate the average firing rate of areas that satisfy two conditions: (i) the area shows persistent activity before inhibition and (ii) the area does not receive inhibitory input. The ratio between such average firing rate after inhibition and before inhibition is used to quantify the overall effect of inhibition. If the ratio is lower than 100%, this suggests that inhibiting certain area(s) disrupts the maintenance of the attractor state. Note that the inhibition effect is typically not very strong, and only in rare cases, inhibition of a single area leads to loss of activity of other areas (*Figure 7B and C*). To quantify such differences, we use a threshold of 10% to differentiate them. We will use (relatively) 'weak inhibition effect' and 'strong inhibition effect' to refer to them afterward.

We used the three measures to classify areas into four types (*Figure 7D*): (i) inhibition effect during delay period, (ii) inhibition effect during stimulus period, and (iii) delay-period firing rate. Areas with strong inhibition effect during stimulus period are classified as input areas; areas with strong inhibition effect during delay period and strong delay-period firing rate are classified as core areas; areas with weak inhibition effect during delay period but strong firing rate are classified as readout areas; and areas with weak inhibition effect during delay period and weak firing rate during delay period are classified as nonessential areas.

As an extension of the single-area inhibition study, we focus on the role of readout areas. A pair of readout areas is randomly chosen and inhibited during the delay period under a similar protocol as the single-area inhibition study. The inhibition effect, that is, the decrement of the delay-period firing rates of other noninhibited areas, is first quantified for each inhibition pair $(A_i, A_j)$. Next, the inhibition effect is averaged one more time for each area $A_i$ across all inhibition pairs that includes the area $((A_i, A_j)$, where $j \neq i)$. An analogous procedure is performed for triplets and quadruplets of readout areas. Additionally, we also calculate the mean inhibition effect between pair of areas, which are both selected from core areas, both selected from readout areas, or we chose one area from core areas, one area from readout areas.

## Simulation of multiple attractors

Multiple attractors coexist in the network and its properties, and number depends on the connectivity and dynamics of each node. In this study, we did not try to capture all the possible attractors in the network, but rather compare the number of attractors for different networks. Here we briefly describe the protocol used to identify multiple attractors in the network. We first choose $k$ areas and then generate a subset of areas as the stimulation areas. We cover all possible subsets, which means we run $2^k$ simulations in total. The external stimulus is given to all areas in the subset simultaneously with same strength and duration. The delay period activity is then quantified using a similar protocol as the standard simulation protocol. The selection of $k$ areas corresponds to a qualitative criterion. First, we choose the areas with small PV fraction or high hierarchy since these areas are more likely to show persistent activity. Second, the number of possible combination grows exponentially as we increase $k$, and if we use $k = 43$, the number of combinations is around 8.8e+12, which is beyond our simulation power. As a trade-off between the simulation power and coverage of areas, we choose $k$

= 18, which correspond to 2.6e+5 different combinations of stimulation. For each parameter setting, we run 2.6e+5 simulations to capture possible attractor patterns. For each attractor pattern, a binary vector is generated by thresholding delay firing rate using a firing rate threshold of 5 Hz. An attractor pattern is considered distinct if and only if the binary vector is different from all identified attractors. In this way, we can identify different attractors in the simulation. We also apply same simulation pipeline to identify attractors for different parameters. Specifically we change the long-range connectivity strength $\mu_{EE}$ and local excitatory connections $g_{E,self}$.

## Acknowledgements

We thank Daniel P Bliss and Ulises Pereira for support with analysis tools at the beginning of the project, and members of the Wang Lab at New York University for discussions related to the project.

## Additional information

### Funding

| Funder | Grant reference number | Author |
|---|---|---|
| National Institutes of Health | R01MH062349 | Xiao-Jing Wang |
| Office of Naval Research | N00014 | Xiao-Jing Wang |
| National Science Foundation | NeuroNex grant | Xiao-Jing Wang |
| Simons Foundation | 543057SPI | Xiao-Jing Wang |
| National Institutes of Health | U19NS123714 | Jorge Jaramillo |
| Biotechnology and Biological Sciences Research Council | BB/X013243/1 | Sean Froudist-Walsh |
| University of Bristol | Neuroscience of Mental Health Award | Sean Froudist-Walsh |
| National Science Foundation | 2015276 | Xiao-Jing Wang |

The funders had no role in study design, data collection and interpretation, or the decision to submit the work for publication.

### Author contributions

Xingyu Ding, Conceptualization, Resources, Data curation, Software, Formal analysis, Validation, Investigation, Visualization, Methodology, Writing – original draft, Writing – review and editing; Sean Froudist-Walsh, Software, Supervision, Methodology, Writing – original draft, Writing – review and editing; Jorge Jaramillo, Supervision, Methodology, Writing – original draft, Writing – review and editing; Junjie Jiang, Methodology, Writing – review and editing; Xiao-Jing Wang, Conceptualization, Resources, Supervision, Funding acquisition, Validation, Methodology, Writing – review and editing

### Author ORCIDs

Xingyu Ding (ID) http://orcid.org/0000-0002-6614-5195
Sean Froudist-Walsh (ID) http://orcid.org/0000-0003-4070-067X
Jorge Jaramillo (ID) http://orcid.org/0000-0002-6666-899X
Junjie Jiang (ID) https://orcid.org/0000-0003-2930-7770
Xiao-Jing Wang (ID) http://orcid.org/0000-0003-3124-8474

### Decision letter and Author response

Decision letter https://doi.org/10.7554/eLife.85442.sa1
Author response https://doi.org/10.7554/eLife.85442.sa2

## Additional files

### Supplementary files
• MDAR checklist

### Data availability

The current manuscript is predominantly a computational study. Consequently, no new data was generated. However, the modeling code that underpins our findings has been made publicly available on GitHub, accessible via https://github.com/XY-DIng/mouse_dist_wm (copy archived at *Ding, 2024*).

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
