## [Editor Report]

This valuable modeling study helps to elucidate the conditions under which interactions across brain regions support working memory. Convincing evidence underscores the importance of not merely the strength and density of long-range connections, but also the cell type specificity of such connections, and solid results indicate that the density of inhibitory neurons, together with placement in a cortical hierarchy, plays a role in how strongly or weakly a brain region displays persistent activity. This work will be of interest to modelers studying the neural basis of working memory, as well as to neuroscientists interested in how global brain interactions shape the patterns of brain activity observed during working memory.

---

## [Decision Letter]

.

**Decision letter after peer review:**

Thank you for submitting your article "Predicting distributed working memory activity in a large-scale mouse brain: the importance of the cell type-specific connectome" for consideration by *eLife*. Your article has been reviewed by 3 peer reviewers, one of whom is a member of our Board of Reviewing Editors, and the evaluation has been overseen Laura Colgin as the Senior Editor. The reviewers have opted to remain anonymous.

Essential revisions:

Overall, the reviewers found the manuscript interesting, but its major claim regarding PV density gradient was incompletely supported by the evidence shown, and the novelty of some of the other important findings was questioned. As a result, the manuscript would likely have to be significantly strengthened upon revision in order to be acceptable for publication in *eLife* (under *eLife*'s old review model, which is how this manuscript was submitted).

1) The manuscript presents at least 3 potential contributing factors to a spatial gradient of persistent activity: hierarchical pattern of connections; PV cell density gradient; counter-stream inhibitory bias (CIB). A principal claim in the manuscript is that the PV cell density gradient is a key factor. But this was not directly demonstrated. The manuscript would be much stronger if the authors directly tested which factor, or which combination of factors, are necessary for a spatial gradient of persistent activity: that is, are hierarchical connections alone, without PV gradient or CIB, sufficient? Are hierarchical connections plus PV gradient, without CIB, sufficient? Are hierarchical connections plus CIB, without PV gradient, sufficient? Etc. It seems the authors could relatively easily test these and directly demonstrate which are the critical ones (including directly testing whether the presence or absence of the PV density gradient is necessary). As pointed out in reviewer 2's Major point 3, how the loop strength results depend on these various factors should also be tested (e.g., are hierarchical organization and CIB sufficient for the results.).

Furthermore, it is essential that the authors look across multiple sensory input modalities before drawing any general conclusions (for example, supplementary data has a version of the task in which sensory inputs are auditory, but then the key analyses upon which the hierarchy claims are based were omitted for this auditory version).

2) A clarification of the graph theoretic measure proposed here should be made. The abstract should make it clear that it simply involves taking into account sign, not only strength, of connections, so readers know what is meant by it, and the word "novel" should be removed – taking into account sign is not a novel concept.

3) The third major point described as a finding in the discussion, that when local recurrent connections are insufficient to support WM, then long-range recurrent interactions between regions are needed, should not be described as a finding of the model, since it seems a logical necessity.

4) Given doubts about several of the critical findings, one aspect that could strengthen the paper enough to be acceptable for *eLife* (old model) would be a substantial strengthening of the mouse-monkey comparison, which would be of interest to many readers and which this group is uniquely qualified to make. Clarification of what is/is not needed to explain the difference in results across the two species would be useful in trying to compare the two. Can the authors use modeling to demonstrate which specific feature(s) are critical for explaining the differences between the two species? Generally speaking, for the discussion: what do the authors see as the prominent similarities or differences across the two species that would be useful to highlight to the community when comparing studies across them?

5) Please describe and discuss in more detail the areas identified as core areas and those identified as readout areas. i.e. why is gustatory part of the core area for visual WM, and why MOs is not a core area but a readout area? If the task used in the model is a visual delayed response task, the difference in task demands cannot explain the discrepancy between the model results and what is concluded from the ALM literature.

6) How redundancy across regions involved in supporting WM interacts with the definition of core vs readout regions. should be discussed. (See major point 2 from reviewer 1)

*Reviewer #1 (Recommendations for the authors):*

Reframing the conclusions around hierarchical position (and the excitatory and inhibitory patterns of connectivity related to the hierarchy that are assumed in the model), and clarifying what is conceptually novel about taking into account sign of connections or stating that long-range connections are needed if short-range connections are insufficient would be very helpful.

*Reviewer #2 (Recommendations for the authors):*

Given that the main results do not seem particularly strong in the current manuscript, it seems like the authors should consider other directions that might be of more interest. For example, I was intrigued by the Discussion about whether ALM is a readout area rather than a core area, and how that might influence the prominent previous work about the role of this area in working memory (although I'm somewhat concerned that, more generally, lesioning one area alone might have minimal effect, and lesioning a different area alone might also have minimal effect, but lesioning both together might lead to a big effect; with respect to the ALM work, it seemed that something like this might be going on with unilateral versus bilateral lesions and I'm not sure if the current model takes that into account). Separately, I'm not sure if there are significant results to put greater emphasis on about multiple areas being involved in working memory, if this differs from what is found in the monkey mesoscale connectome models, or if there are other enough other novel differences between mouse and monkey that could be emphasized.

Clarification points:

1) Fonts on brain areas are tiny/unreadable, as well as on figure insets and axis labels throughout the paper, including the supplemental. The yellow color on the white background is very hard to read.

2) It is confusing to try to distinguish in the text when the PV cell fraction is/is not normalized. This should be clarified throughout the main text and figure captions, as well as justifying when it is appropriate to use the normalized measure.

3) Figure 1 – Supplement 1C. Would be helpful to plot this in the same order as panel B.

*Reviewer #3 (Recommendations for the authors):*

1. The potential significance of the results is at times hindered by the lack of details/clarity in the methods/results description. For example, important concepts such as cortical hierarchy are only explained in the Method section, but not in the Results section. Other rather complex concepts, such as counterstream inhibitory bias (lines 112-114), can also be more clearly described, even if it has already been covered by previous papers.

2. Figure 5 can be better annotated. For example, it is difficult to confirm (from legend) whether Figure 5B is related to sensory-period inhibition or delay-period inhibition. There should be a more explicit color code for the different categories in Figure 5D, which I only now realise matches the color code in Figure 5A.

3. Some of the results from Figure 5 are very loosely alluded to in the main text (e.g., in lines 250-251, which did not name the readout regions).

4. Although the thalamic-cortical model is quite interesting, it does feel rather separate from the rest of the study. Does this part really belong to this paper? Maybe the authors could link it better with the other main results.

5. In general, the paper could be strengthened with more clear writing and more coherent organization. As an experimentalist, I find it difficult to understand the technical details of this paper. Mejias and Wang, 2022 are a good example of very similar work suitable for a more general audience.

---

## [Author Response]

Essential revisions:Overall, the reviewers found the manuscript interesting, but its major claim regarding PV density gradient was incompletely supported by the evidence shown, and the novelty of some of the other important findings was questioned. As a result, the manuscript would likely have to be significantly strengthened upon revision in order to be acceptable for publication in eLife (under eLife's old review model, which is how this manuscript was submitted).

We would like to thank the reviewers for their constructive feedback. We have addressed their specific concerns and included 2 additional main figures and 1 supplementary figures on the contributions of PV gradient and hierarchy, 3 supplementary figures on other issues raised in the Essential Revisions, a thorough revision of some of the claims made in the manuscript. In addition, we addressed individual reviewers’ concerns. We reiterate our intention to continue with *eLife*’s old review model. We first provide a summary of the most important changes:

First, we produced new simulations to directly evaluate the importance of counter-stream inhibitory bias (CIB) and PV cell density in the generation of working memory patterns across the thalamocortical network. Second, we examined how several graph-theoretic measures predict neural dynamics and address the interplay between these measures and the presence or absence of PV gradient and CIB. Third, we explored different parameter regimes for delay-period activity in the large-scale model and explained the contributions of long-range connections to both working memory patterns as well as to the stability of the baseline state. Fourth, we produced new simulations where multiple readout areas are inhibited and evaluated the effect on persistent activity at the network level. Finally, we compared rodent and primate neural circuitry and how large-scale models such as this one can predict and/or explain salient differences between these species. We also rewrote the Abstract and avoided gratuitous use of the words ‘novel/novelty’.

1) The manuscript presents at least 3 potential contributing factors to a spatial gradient of persistent activity: hierarchical pattern of connections; PV cell density gradient; counter-stream inhibitory bias (CIB). A principal claim in the manuscript is that the PV cell density gradient is a key factor. But this was not directly demonstrated. The manuscript would be much stronger if the authors directly tested which factor, or which combination of factors, are necessary for a spatial gradient of persistent activity: that is, are hierarchical connections alone, without PV gradient or CIB, sufficient? Are hierarchical connections plus PV gradient, without CIB, sufficient? Are hierarchical connections plus CIB, without PV gradient, sufficient? Etc. It seems the authors could relatively easily test these and directly demonstrate which are the critical ones (including directly testing whether the presence or absence of the PV density gradient is necessary). As pointed out in reviewer 2's Major point 3, how the loop strength results depend on these various factors should also be tested (e.g., are hierarchical organization and CIB sufficient for the results.).Furthermore, it is essential that the authors look across multiple sensory input modalities before drawing any general conclusions (for example, supplementary data has a version of the task in which sensory inputs are auditory, but then the key analyses upon which the hierarchy claims are based were omitted for this auditory version).

To address this important comment, we varied the parameters that characterized PV gradients and CIB and compared them to the model used in the manuscript that includes both PV gradient and CIB. Note also that the hierarchy only affects the model via the CIB (equations 21-23). Therefore, we focused our analyses on the PV gradient and the CIB. First, we evaluated the importance of the PV gradient in determining spatially-patterned activity across the cortex. We replaced the PV gradient, which varies across areas, with a constant value across areas. We found that the abrupt transition in firing rate activity at a particular area, a hallmark of the large-scale model, remains. However, the number of cortical areas with high firing rate (i.e., higher than baseline) is diminished compared to the original model.

To address this important comment, we varied the parameters that characterized PV gradients and CIB and compared them to the model used in the manuscript that includes both PV gradient and CIB. Note also that the hierarchy only affects the model via the CIB (equations 21-23). Therefore we focused our analyses on the PV gradient and the CIB. First, we evaluated the importance of the PV gradient in determining spatially-patterned activity across the cortex. We replaced the PV gradient, which varies across areas, with a constant value across areas. We found that the abrupt transition in firing rate activity at a particular area, a hallmark of the large-scale model, remains. However, the number of cortical areas with high firing rate (i.e., higher than baseline) is diminished compared to the original model.

We added the following text to the Results section:

“We explored the potential contributions of PV gradients and CIB in determining spatially patterned activity across the cortex. […] In summary, we have shown that distinct local and long-range inhibitory mechanisms shape the pattern of working memory activity and stability of the baseline state.”

Based on our new simulations and results, we added the following paragraph to the Discussion section:

“This preferential targeting of feedback projections serves to stabilize the otherwise excitable activity of sensory areas (Mejias et al., 2021), and is consistent with recent experiments that report long-range recruitment of GABAergic neurons in early sensory areas (Campagnola et al., 2022, Li et al., 2021, Naskar et al., 2021). […] Given that working memory is a fundamental cognitive function observed across many individual brains with anatomical differences, the inclusion of multiple inhibitory mechanisms that allow for connectivity variations might confer evolutionary advantages.”

2) A clarification of the graph theoretic measure proposed here should be made. The abstract should make it clear that it simply involves taking into account sign, not only strength, of connections, so readers know what is meant by it, and the word "novel" should be removed – taking into account sign is not a novel concept.

We have rewritten this section of the abstract and removed the word ‘novel. In the original manuscript, we proposed a graph theoretic measure to elucidate the relationship between structural connectivity and large-scale neural dynamics. More specifically, we evaluated the extent to which graph theory metrics first, predict the firing rate activity patterns during the delay period and second, predict whether a given cortical area is core or not. While the inclusion of sign is an important aspect of this graph-theoretic measure – and not novel – our measure includes other important aspects that we did not communicate with sufficient clarity. Our graph-theoretic measure includes: (i) the strength of both local and long-range connections, and (ii) the weighted contributions of CIB and PV. Standard graph theory measures treat each node identically. We show that accounting for variation across nodes (e.g. in PV density) one can improve the classification of working memory core areas. More generally, accounting for variation in local node properties could enhance graph-theory analyses in neuroscience, which may have applications in other fields. Following the reviewers’ suggestions and to better illustrate the above contributions, we have implemented two other measures for comparison: (a) a loop-strength measure that adds a ‘sign’ without further modification, and (b) a loop strength measure that takes hierarchical information – and not PV information- into account. We found that ‘raw’ input strength cannot predict firing rate during the delay period of a sensory working memory task. As soon as we include hierarchical information that distinguishes excitatory from inhibitory feedback, this measure can be used to predict delay firing rate. On the other hand, the prediction of the core areas greatly depends on cell-type specificity: the sign-only and ‘no-PV’ measures do not reliably predict whether an area is a core area or not, highlighting the importance of cell-type specific connectivity measures.

We hope that this explanation provides a clearer understanding of the graph theoretic measure we have proposed. We have also modified the abstract to reflect the importance of the measure while removing references to its novelty. We added the following paragraph to the Results section and an accompanying supplementary figure:

“To better demonstrate our cell type-specific connectivity measures, we have implemented two other measures for comparison: (a) a loop-strength measure that adds a ‘sign’ without further modification, and (b) a loop strength measure that takes hierarchical information – and not PV information- into account. These two graph-theoretic measures can be used to predict delay firing rate during a sensory working memory task, thus highlighting the importance of hierarchical information, which distinguishes excitatory from inhibitory feedback (Figure 6-supplement 3). On the other hand, the prediction of the core areas greatly depends on cell-type specificity: the sign-only and ‘no-PV’ mechanisms do not reliably predict whether an area is a core area or not, especially in the case of calculating with length 3 loops, demonstrating the importance of cell-type specific connectivity measures. (Figure 8 – supplement 2).”

3) The third major point described as a finding in the discussion, that when local recurrent connections are insufficient to support WM, then long-range recurrent interactions between regions are needed, should not be described as a finding of the model, since it seems a logical necessity.

We thank the reviewers for highlighting this point as we don’t want to overemphasize a potentially trivial aspect. There are two main points that our simulations have revealed regarding long-range recurrent interactions: First, the way long-range recurrent interactions affect neural dynamics depends not only on the existence of excitatory projections per se, but also on the target neurons’ cell type. Thus, for some cortical areas afferent long-range excitatory connection promotes memory-related activity while for some others, e.g., early sensory areas, it does not. Second, distributed persistent activity in our model exists for different parameter regimes that depend on long-range excitatory interactions. These regimes are functionally distinct and can be identified via a perturbation analysis. For example, in one regime some areas exhibit independent persistent activity, i.e., local recurrent interactions are sufficient to sustain a memory state for these areas, while others do not. In another regime, none of the areas can sustain a memory state without receiving long-range input. Moreover, in this regime CIB is not required for the existence of distributed persistent activity patterns. These two regimes are functionally distinct in terms of robustness as well as in the number of attractors that they can sustain.

We added the following paragraph to the Discussion

“To conclude, the manner in which long-range recurrent interactions affect neural dynamics depends not only on the existence of excitatory projections per se, but also on the target neurons’ cell type. Thus, for some cortical areas afferent long-range excitatory connections promote working memory-related activity while for some others, e.g., early sensory areas, it does not. Moreover, the existence of long-range interactions is consistent with potentially distinct dynamical regimes. For example, in one regime some areas exhibit independent persistent activity, i.e., local recurrent interactions are sufficient to sustain a memory state for these areas, while others do not. In this regime CIB is not required for the existence of distributed persistent activity patterns. In another regime, none of the areas can sustain a memory state without receiving long-range input. These two regimes are functionally distinct in terms of their robustness to perturbation as well as in the number of attractors that they can sustain. These regimes may be identified via perturbation analysis in future experimental and theoretical work.”

4) Given doubts about several of the critical findings, one aspect that could strengthen the paper enough to be acceptable for eLife (old model) would be a substantial strengthening of the mouse-monkey comparison, which would be of interest to many readers and which this group is uniquely qualified to make. Clarification of what is/is not needed to explain the difference in results across the two species would be useful in trying to compare the two. Can the authors use modeling to demonstrate which specific feature(s) are critical for explaining the differences between the two species? Generally speaking, for the discussion: what do the authors see as the prominent similarities or differences across the two species that would be useful to highlight to the community when comparing studies across them?

Comparing modeling results across animal species is an important research direction that has not been sufficiently explored in general, and with regards to distributed working memory in particular. While a detailed comparison based on simulations is outside of the scope of the current study, we can draw inferences based on previous large-scale modeling work based on the macaque connectome (e.g., Chaudhuri et al., 2015, Mejias and Wang 2022, Froudist-Walsh et al., 2021) as well as recent anatomical and physiological literature that directly relates to our modeling approach.

For early sensory areas, for example, there may be important differences across species. Inhibition from either the PV gradient or CIB may be important to maintain the stability of an otherwise highly excitable cortical area. In our model, we used the PV gradient reported in Kim et al., 2017 to constrain local and long-range circuits that provide inhibition to and maintain stability in early sensory areas. Along these lines, we predict that local recurrency in the mouse early sensory areas is higher than in the primate. Consistent with this claim, spine density in layer 3 pyramidal neurons is higher in mouse than macaque V1 (Figure 5A in Gilman et al.,. 2017), and the total number of excitatory and inhibitory synapses and spine density in layer 2/3 of mouse V1 – as quantified by electron microscopy- is higher than primate V1, S1, and LIP (Figure 1A in Wildenberg et al., 2021).

We have added the following to the Discussion section.

“More recently, large-scale neural circuit models have been developed specifically to reproduce neural activity during cognitive tasks (Mejias et al., 2022, Froudist-Walsh et al., 2021, Klatzmann et al., 2022). […] Although there is some evidence for similar inhibitory gradients in humans (Burt et al., 2018) and macaque (Torres-Gomez et al., 2020), the computational consequences of differences across species remain to be established.”

5) Please describe and discuss in more detail the areas identified as core areas and those identified as readout areas. i.e. why is gustatory part of the core area for visual WM, and why MOs is not a core area but a readout area? If the task used in the model is a visual delayed response task, the difference in task demands cannot explain the discrepancy between the model results and what is concluded from the ALM literature.

Some of the areas that were identified as core areas in our model have been widely studied in other tasks. For example, the gustatory area exhibits delay-period preparatory activity in a taste-guided decision-making task and inhibition of this area during the delay period impairs behavior in this task (Vincis et al., 2020). Our graph-theory metric, the ‘cell-type specific loop strength’, measures the strength of excitatory loops weighted by cell type and explains why the gustatory and other areas were found to be core. More generally, we conclude that there is a core substrate for sensory working memory, largely independent of the modality.

With the cell-type specific loop strength metric, we also concluded that MOs is a readout area and not a core area. This finding may be surprising considering previous studies that have shown this area to participate in short-term memory, planning, and movement execution during a memory-guided response task (Svoboda and Li, 2018, Inagaki et al., 2022). This task has shown to engage, not only ALM, but a distributed subcortical-cortical network that includes the thalamus, basal ganglia and cerebellum (Svoboda and Li, 2018). We note that in this version of the memory-guided response task, short-term memory is conflated with movement preparation. In our task, we proposed to study the maintenance of sensory information, without regards to the movement preparation component. It is for this context that we found that MOs is not a core area. Indeed, there is evidence for a differential engagement of cortical networks depending on the task design (Jonikaitis et al., 2023) and on effectors (Kubanek and Synder, 2015).

We would like to emphasize that readout areas are not less important than core areas as readout areas are expected to be strongly coupled to behavior. Thus, it is difficult to dissociate readout from core areas based on behavior alone, which justifies our inhibition-based approach. Importantly, readout areas may be particularly crucial for behaviors once the memory is no longer needed, e.g., during movement execution (Li et al., 2015; Svoboda and Li, 2018).

We have added the following paragraph to the Discussion:

“Some of the areas that were identified as core areas in our model have been widely studied in other tasks. […] Indeed, there is evidence for a differential engagement of cortical networks depending on the task design (Jonikaitis et al. 2023) and on effectors (Kubanek and Snyder 2015).”

6) How redundancy across regions involved in supporting WM interacts with the definition of core vs readout regions. should be discussed. (See major point 2 from reviewer 1)

We initially defined a core area for working memory maintenance as a cortical area that, first, exhibits persistent activity, and second, removal of this area (e.g., experimentally via a lesion or opto-inhibition) significantly affects persistent activity in other areas. Following reviewer 1’s comments and insightful thought experiment, we proceeded with inhibiting two, three, and four readout areas concurrently. Our analysis demonstrated that the classification of areas into e.g., core vs readout follows a hierarchy: second-order core areas are those areas that have a strong inhibition effect when inhibiting pairs of areas; the third orders are triplets, and the fourth orders are quadruplets. Thus, readout areas are not equivalent to each other. Moreover, the effects of silencing pairs, triplets and quadruplets of readout areas remain smaller than those seen after silencing single core areas listed above. Thus, the pattern of persistent activity is more sensitive to perturbations of core areas, which underscores the classification of some cortical areas into core vs readout. We have added two new paragraphs in the Results section, a new supplementary figure 7, and corresponding description in the Methods section.

We added the following paragraphs to the Results section

“We have defined a core area for working memory maintenance as a cortical area that, first, exhibits persistent activity, and second, removal of this area (e.g., experimentally via a lesion or opto-inhibition) significantly affects persistent activity in other areas. […] Thus, the pattern of persistent activity is more sensitive to perturbations of core areas, which underscores the classification of some cortical areas into core vs readout.”

Reviewer #1 (Recommendations for the authors):Reframing the conclusions around hierarchical position (and the excitatory and inhibitory patterns of connectivity related to the hierarchy that are assumed in the model), and clarifying what is conceptually novel about taking into account sign of connections or stating that long-range connections are needed if short-range connections are insufficient would be very helpful.

We have included this in our revision, thank you.

Reviewer #2 (Recommendations for the authors):Given that the main results do not seem particularly strong in the current manuscript, it seems like the authors should consider other directions that might be of more interest. For example, I was intrigued by the Discussion about whether ALM is a readout area rather than a core area, and how that might influence the prominent previous work about the role of this area in working memory (although I'm somewhat concerned that, more generally, lesioning one area alone might have minimal effect, and lesioning a different area alone might also have minimal effect, but lesioning both together might lead to a big effect; with respect to the ALM work, it seemed that something like this might be going on with unilateral versus bilateral lesions and I'm not sure if the current model takes that into account). Separately, I'm not sure if there are significant results to put greater emphasis on about multiple areas being involved in working memory, if this differs from what is found in the monkey mesoscale connectome models, or if there are other enough other novel differences between mouse and monkey that could be emphasized.

We thank the reviewer for these comments. We have addressed them in our most recent version of the manuscript, while some limitations and future work were included in the Discussion.

Clarification points:1) Fonts on brain areas are tiny/unreadable, as well as on figure insets and axis labels throughout the paper, including the supplemental. The yellow color on the white background is very hard to read.

We improved the readability of the Figures and captions throughout.

2) It is confusing to try to distinguish in the text when the PV cell fraction is/is not normalized. This should be clarified throughout the main text and figure captions, as well as justifying when it is appropriate to use the normalized measure.

We now use PV cell fraction throughout, which depends on a normalized PV density. Results and methods were updated.

3) Figure 1 – Supplement 1C. Would be helpful to plot this in the same order as panel B.

We have now used the same x axis for both plots.

Reviewer #3 (Recommendations for the authors):1. The potential significance of the results is at times hindered by the lack of details/clarity in the methods/results description. For example, important concepts such as cortical hierarchy are only explained in the Method section, but not in the Results section. Other rather complex concepts, such as counterstream inhibitory bias (lines 112-114), can also be more clearly described, even if it has already been covered by previous papers.

We now introduced the concepts of cortical hierarchy and CIB in the Results section.

2. Figure 5 can be better annotated. For example, it is difficult to confirm (from legend) whether Figure 5B is related to sensory-period inhibition or delay-period inhibition. There should be a more explicit color code for the different categories in Figure 5D, which I only now realise matches the color code in Figure 5A.

We have updated the figure and captions.

3. Some of the results from Figure 5 are very loosely eluded to in the main text (e.g., in lines 250-251, which did not name the readout regions).

We have now added a more comprehensive description of readout vs core areas in the Results and Discussion.

4. Although the thalamic-cortical model is quite interesting, it does feel rather separate from the rest of the study. Does this part really belong to this paper? Maybe the authors could link it better with the other main results.

We added some sentences in the Discussion to better link the results of the thalamocortical model to our other results, including limitations in the model and further work.

5. In general, the paper could be strengthened with more clear writing and more coherent organization. As an experimentalist, I find it difficult to understand the technical details of this paper. Mejias and Wang, 2022 are a good example of very similar work suitable for a more general audience.

Similar to points 1 and 4 above, in our revision we have introduced necessary concepts to improve the logical flow and readability of the manuscript.